# When Does Confidence-Based Cascade Deferral Suffice?

**Wittawat Jitkrittum**      **Neha Gupta**      **Aditya Krishna Menon**
**Harikrishna Narasimhan**      **Ankit Singh Rawat**      **Sanjiv Kumar**

Google Research, New York

{wittawat, nehagup, adityakmenon, hnarasimhan, ankitsrawat, sanjivk}@google.com

## Abstract

Cascades are a classical strategy to enable inference cost to vary *adaptively* across samples, wherein a sequence of classifiers are invoked in turn. A *deferral rule* determines whether to invoke the next classifier in the sequence, or to terminate prediction. One simple deferral rule employs the *confidence* of the current classifier, e.g., based on the maximum predicted softmax probability. Despite being oblivious to the structure of the cascade — e.g., not modelling the errors of downstream models — such confidence-based deferral often works remarkably well in practice. In this paper, we seek to better understand the conditions under which confidence-based deferral may fail, and when alternate deferral strategies can perform better. We first present a theoretical characterisation of the optimal deferral rule, which precisely characterises settings under which confidence-based deferral may suffer. We then study *post-hoc* deferral mechanisms, and demonstrate they can significantly improve upon confidence-based deferral in settings where (i) downstream models are *specialists* that only work well on a subset of inputs, (ii) samples are subject to label noise, and (iii) there is distribution shift between the train and test set.

## 1 Introduction

Large neural models with several billions of parameters have shown considerable promise in challenging real-world problems, such as language modelling [57, 58, 3, 59] and image classification [15, 20]. While the quality gains of these models are impressive, they are typically accompanied with a sharp increase in inference time [6, 67], thus limiting their applicability. *Cascades* offer one strategy to mitigate this [77, 75, 62, 24], by allowing for faster predictions on "easy" samples. In a nutshell, cascades involve arranging multiple models in a sequence of increasing complexities. For any test input, one iteratively applies the following recipe, starting with the first model in the sequence: execute the current model, and employ a *deferral rule* to determine whether to invoke the next model, or terminate with the current model's prediction. One may further combine cascades with ensembles to significantly improve accuracy-compute trade-offs [22, 80].

A key ingredient of cascades is the choice of deferral rule. The simplest candidate is to defer when the current model's *confidence* in its prediction is sufficiently low. Popular confidence measures include the maximum predictive probability over all classes [80], and the entropy of the predictive distribution [31]. Despite being oblivious to the nature of the cascade — e.g., not modelling the errors of downstream models — such *confidence-based deferral* works remarkably well in practice [81, 23, 21, 80, 47, 49]. Indeed, it has often been noted that such deferral can perform on-par with

37th Conference on Neural Information Processing Systems (NeurIPS 2023).

more complex modifications to the model training [23, 21, 36]. However, the reasons for this success remain unclear; further, it is not clear if there are specific practical settings where confidence-based deferral may perform poorly [50].

In this paper, we initiate a systematic study of the potential limitations of confidence-based deferral for cascades. Our findings and contributions are:

(i) We establish a novel result characterising the theoretically optimal deferral rule (Proposition 3.1), which for a two-model cascade relies on the confidence of both model 1 and model 2.

(ii) In many regular classification tasks where model 2 gives a consistent estimate of the true posterior probability, confidence-based deferral is highly competitive. However, we show that in some settings, confidence-based deferral can be significantly sub-optimal both in theory and practice (§4.1). This includes when (1) model 2's error probability is highly non-uniform across samples, which can happen when model 2 is a *specialist* model, (2) labels are subject to noise, and (3) there is distribution shift between the train and test set.

(iii) Motivated by this, we then study a series of *post-hoc* deferral rules, that seek to mimic the form of the optimal deferral rule (§4.2). We show that post-hoc deferral can significantly improve upon confidence-based deferral in the aforementioned settings.

To the best of our knowledge, this is the first work that precisely identifies specific practical problems settings where confidence-based deferral can be sub-optimal for cascades. Our findings give insights on when it is appropriate to deploy a confidence-based cascade model in practice.

## 2 Background and Related Work

Fix an instance space $\mathcal{X}$ and label space $\mathcal{Y} = [L] \doteq \{1, 2, \ldots, L\}$, and let $\mathbb{P}$ be a distribution over $\mathcal{X} \times \mathcal{Y}$. Given a training sample $S = \{(x_n, y_n)\}_{n \in [N]}$ drawn from $\mathbb{P}$, multi-class classification seeks a *classifier* $h \colon \mathcal{X} \to \mathcal{Y}$ with low *misclassification error* $R(h) = \mathbb{P}(y \neq h(x))$. We may parameterise $h$ as $h(x) = \operatorname{argmax}_{y' \in \mathcal{Y}} f_{y'}(x)$ for a *scorer* $f \colon \mathcal{X} \to \mathbb{R}^L$. In neural models, one expresses $f_y(x) = w_y^\top \Phi(x)$ for class weights $w_y$ and embedding function $\Phi$. It is common to construct a *probability estimator* $p \colon \mathcal{X} \to \Delta^L$ using the softmax transformation, $p_y(x) \propto \exp(f_y(x))$.

### 2.1 Cascades and Deferral Rules

Conventional neural models involve a fixed inference cost for any test sample $x \in \mathcal{X}$. For large models, this cost may prove prohibitive. This has motivated several approaches to *uniformly* lower the inference cost for *all* samples, such as architecture modification [41, 76, 31, 78], stochastic depth [30, 19], network sparsification [44], quantisation [32], pruning [42], and distillation [5, 29, 61]. A complementary strategy is to *adaptively* lower the inference for *"easy"* samples, while reserving the full cost only for "hard" samples [26, 86, 66, 55, 72, 81, 31, 68, 83, 17, 45, 85, 82].

Cascade models are a classic example of adaptive predictors, which have proven useful in vision tasks such as object detection [77], and have grown increasingly popular in natural language processing [47, 75, 38, 14]. In the vision literature, cascades are often designed for binary classification problems, with lower-level classifiers being used to quickly identify negative samples [4, 84, 63, 10, 13, 70]. A cascade is composed of two components:

(i) a collection of base models (typically of non-decreasing inference cost)

(ii) a *deferral rule* (i.e., a function that decides which model to use for each $x \in \mathcal{X}$).

For any test input, one executes the first model, and employs the deferral rule to determine whether to terminate with the current model's prediction, or to invoke the next model; this procedure is repeated until one terminates, or reaches the final model in the sequence. Compared to using a single model, the goal of the cascade is to offer comparable predictive accuracy, but lower average inference cost.

While the base models and deferral rules may be trained jointly [74, 36], we focus on a setting where the base models are *pre-trained and fixed*, and the goal is to train only the deferral rule. This setting is practically relevant, as it is often desirable to re-use powerful models that involve expensive training procedures. Indeed, a salient feature of cascades is their ability to leverage off-the-shelf models and simply adjust the desired operating point (e.g., the rate at which we call a large model).

## 2.2 Confidence-based Cascades

A simple way to define a deferral rule is by thresholding a model's *confidence* in its prediction. While there are several means of quantifying and improving such confidence [69, 25, 37, 34], we focus on the *maximum predictive probability* $\gamma(x) \doteq \max_{y'} p(y' \mid x)$. Specifically, given $K$ trained models, a confidence-based cascade is formed by picking the first model that whose confidence $\gamma(x)$ is sufficiently high [80, 60]. This is made precise in Algorithm 1.

Forming a cascade in this manner is appealing because it does not require retraining of any models or the deferral rule. Such a post-hoc approach has been shown to give good trade-off between accuracy and inference cost [80]. Indeed, it has often been noted that such deferral can perform on-par with more complex modifications to the model training [23, 21, 36]. However, the reasons for this phenomenon are not well-understood; further, it is unclear if there are practical settings where confidence-based cascades are expected to underperform.

To address this, we now formally analyse confidence-based cascades, and explicate their limitations.

---

**Algorithm 1** Confidence-based cascades of $K$ classifiers

---

**Input:** $K \geq 2$ classifiers: $p^{(1)}, \ldots, p^{(K)} \colon \mathcal{X} \to \Delta^L$, thresholds $c^{(1)}, \ldots, c^{(K-1)} \in [0,1]$
**Input:** An input instance $x \in \mathcal{X}$

1: **for** $k = 1, 2, \ldots, K-1$ **do**
2:     **if** $\max_{y'} p_{y'}^{(k)}(x) > c^{(k)}$ **then**
3:         Predict class $\hat{y} = \operatorname{argmax}_{y'} p_{y'}^{(k)}(x)$
4:         **break**
5:     **end if**
6: **end for**
7: Predict class $\hat{y} = \operatorname{argmax}_{y'} p_{y'}^{(K)}(x)$

---

# 3 Optimal Deferral Rules for Cascades

In this section, we derive the oracle (Bayes-optimal) deferral rule, which will allow us to understand the effectiveness and limitations of confidence-based deferral (Algorithm 1).

## 3.1 Optimisation Objective

For simplicity, we consider a cascade of $K = 2$ pre-trained classifiers $h^{(1)}, h^{(2)} \colon \mathcal{X} \to \mathcal{Y}$; our analysis readily generalises to cascades with $K > 2$ models (see Appendix F). Suppose that the classifiers are based on probability estimators $p^{(1)}, p^{(2)}$, where for $i \in \{1, 2\}$, $p_{y'}^{(i)}(x)$ estimates the probability for class $y'$ given $x$, and $h^{(i)}(x) \doteq \arg\max_{y'} p_{y'}^{(i)}(x)$. We do not impose any restrictions on the training procedure or performance of these classifiers. We seek to learn a deferral rule $r \colon \mathcal{X} \to \{0, 1\}$ that can decide whether $h^{(1)}$ should be used (with $r(x) = 0$), or $h^{(2)}$ (with $r(x) = 1$).

**Constrained formulation**. To derive the optimal deferral rule, we must first specify a target metric to optimise. Recall that cascades offer a suitable balance between average inference cost, and predictive accuracy. Let us assume without loss of generality that $h^{(2)}$ has higher computational cost, and that invoking $h^{(2)}$ incurs a constant cost $c > 0$. We then seek to find a deferral rule $r$ that maximises the predictive accuracy, while invoking $h^{(2)}$ sparingly. This can be formulated as the following constrained optimisation problem:

$$\min_{r} \mathbb{P}(y \neq h^{(1)}(x), r(x) = 0) + \mathbb{P}(y \neq h^{(2)}(x), r(x) = 1) \quad \text{subject to} \quad \mathbb{P}(r(x) = 1) \leq \tau, \quad (1)$$

for a deferral rate $\tau \in [0, 1]$. Intuitively, the objective measures the misclassification error only on samples where the respective classifier's predictions are used; the constraint ensures that $h^{(2)}$ is invoked on at most $\tau$ fraction of samples. It is straight-forward to extend this formulation to generic cost-sensitive variants of the predictive accuracy [18].

**Equivalent unconstrained risk**. Using Lagrangian theory, one may translate (1) into minimizing an equivalent unconstrained risk. Specifically, under mild distributional assumptions (see e.g., Neyman and Pearson [51]), we can show that for any deferral rate $\tau$, there exists a deferral cost $c \geq 0$ such that solving (1) is equivalent to minimising the following unconstrained risk:

$$R(r; h^{(1)}, h^{(2)}) = \mathbb{P}(y \neq h^{(1)}(x), r(x) = 0) + \mathbb{P}(y \neq h^{(2)}(x), r(x) = 1) + c \cdot \mathbb{P}(r(x) = 1). \quad (2)$$

**Deferral curves and cost-risk curves**. As $c$ varies, one may plot the misclassification error of the resulting cascade as a function of deferral rate. We will refer to this as the *deferral curve*. One may similarly compute the *cost-risk curve* that traces out the optimal cascade risk (2) as a function of $c$. Both these are equivalent ways of assessing the overall quality of a cascade.

In fact, the two curves have an intimate point-line duality. Any point $(\tau, E)$ in deferral curve space – where $\tau$ denotes deferral rate, and $E$ denotes error – can be mapped to the line $(c, c \cdot \tau + E) : c \in [0, 1]$ in cost curve space. Conversely, any point $(c, R)$ in cost curve space – where $c$ denotes cost, and $R$ denotes risk – can be mapped to the line $(d, R - C \cdot d) : d \in [0, 1]$ in deferral curve space. This is analogous to the correspondence between ROC and cost-risk curves in classification [16]. Following prior work [2, 36], our experiments use deferral curves to compare methods.

## 3.2 The Bayes-Optimal Deferral Rule

The *Bayes-optimal* rule, which minimises the risk $R(r; h^{(1)}, h^{(2)})$ over all possible deferral rules $r \colon \mathcal{X} \to \{0, 1\}$, is given below.

**Proposition 3.1.** *Let $\eta_{y'}(x) \doteq \mathbb{P}(y'|x)$. Then, the Bayes-optimal deferral rule for the risk in (2) is:*

$$r^*(x) = \mathbf{1}\left[\eta_{h^{(2)}(x)}(x) - \eta_{h^{(1)}(x)}(x) > c\right], \tag{3}$$

(Proof in Appendix A.) Note that for a classifier $h$, $\eta_{h(x)}(x) = \mathbb{P}(y = h(x)|x)$ i.e., the probability that $h$ gives a correct prediction for $x$. Observe that the randomness here reflects the inherent stochasticity of the labels $y$ for an input $x$, i.e., the aleatoric uncertainty [33]. For the purposes of evaluating the deferral curve, the key quantity is $\eta_{h^{(2)}(x)}(x) - \eta_{h^{(1)}(x)}(x)$, i.e., the *difference in the probability of correct prediction under each classifier*. This is intuitive: it is optimal to defer to $h^{(2)}$ if the expected reduction in misclassification error exceeds the cost of invoking $h^{(2)}$.

The Bayes-optimal deferral rule in (3) is a theoretical construct, which relies on knowledge of the true posterior probability $\eta$. In practice, one is likely to use an approximation to this rule. To quantify the effect of such an approximation, we may consider the *excess risk* or *regret* of an arbitrary deferral rule $r$ over $r^*$. We have the following.

**Corollary 3.2.** *Let $\alpha(x) \doteq \eta_{h_1(x)}(x) - \eta_{h_2(x)}(x) + c$. Then, the excess risk for an arbitrary $r$ is*

$$R(r; h^{(1)}, h^{(2)}) - R(r^*; h^{(1)}, h^{(2)}) = \mathbb{E}_x[(\mathbf{1}(r(x) = 1) - \mathbf{1}(\alpha(x) < 0)) \cdot \alpha(x)].$$

Intuitively, the above shows that when we make deferral decisions that disagree with the Bayes-optimal rule, we are penalised proportional to the difference between the two models' error probability.

## 3.3 Plug-in Estimators of the Bayes-Optimal Deferral Rule

In practice, we may seek to approximate the Bayes-optimal deferral rule $r^*$ in (3) with an estimator $\hat{r}$. We now present several *oracle estimators*, which will prove useful in our subsequent analysis.

**One-hot oracle**. Observe that $\eta_{y'}(x) = \mathbb{E}_{y|x}[\mathbf{1}[y = y']]$. Thus, given a test sample $(x, y)$, one may replace the expectation with the observed label $y$ to yield the ideal estimator $\hat{\eta}_{y'}(x) = \mathbf{1}[y' = y]$. This results in the rule

$$\hat{r}_{01}(x) \doteq \mathbf{1}\left[\mathbf{1}[y = h^{(2)}(x)] - \mathbf{1}[y = h^{(1)}(x)] > c\right]. \tag{4}$$

One intuitive observation is that for high $c$, this rule only defers samples with $y \neq h^{(1)}(x)$ but $y = h^{(2)}(x)$, i.e., *samples where the first model is wrong, but the second model is right*. Unfortunately, this rule is impractical, since it depends on the label $y$. Nonetheless, it serves as an oracle to help understand what one can gain if we knew exactly whether the downstream model makes an error.

**Probability oracle**. Following a similar reasoning as $\hat{r}_{01}$, another estimator is given by

$$\hat{r}_{\text{prob}}(x) \doteq \mathbf{1}\left[p_y^{(2)}(x) - p_y^{(1)}(x) > c\right]. \tag{5}$$

Intuitively $p_y^{(2)}(x)$ can be seen as a label-dependent correctness score of model 2 on an instance $x$.

**Relative confidence**. The above oracles rely on the true label $y$. A more practical plug-estimator is $\hat{\eta}_{h^{(i)}(x)}(x) = \max_{y'} p_{y'}^{(i)}(x)$, which simply uses each model's softmax probabilities. The rationale for this rests upon the assumption that the probability model $p^{(i)}$ is a consistent estimate of the true posterior probability so that $\mathbb{P}(y|x) \approx p_y^{(i)}(x)$ for $i \in \{1, 2\}$, where $(x, y)$ is a labeled example. Thus, $\eta_{h^{(i)}(x)}(x) = \mathbb{P}(h^{(i)}(x)|x) \approx p^{(i)}(h^{(i)}(x)|x) = \max_{y'} p_{y'}^{(i)}(x)$, resulting in the rule

$$\hat{r}_{\text{rel}}(x) \doteq 1\left[\max_{y''} p_{y''}^{(2)}(x) - \max_{y'} p_{y'}^{(1)}(x) > c\right]. \tag{6}$$

Observe that this deferral decision depends on the confidence of *both* models, in contrast to confidence-based deferral which relies only on the confidence of the *first* model.

Note that the above oracles cannot be used directly for adaptive computation, because the second model is invoked on *every* input. Nonetheless, they can inform us about the available headroom to improve over confidence-based deferral by considering the confidence of the downstream model. As shall be seen in §4, these estimators are useful for deriving objectives to train a post hoc deferral rule.

### 3.4 Relation to Existing Work

The two-model cascade is closely connected to the literature on *learning to defer to an expert* [46, 48, 9]. Here, the goal is to learn a base classifier $h^{(1)}$ that has the option of invoking an "expert" model $h^{(2)}$; this invocation is controlled by a deferral rule $r$. Indeed, the risk (2) is a special case of Mozannar and Sontag [48, Equation 2], where the second model is considered to be an "expert". Proposition 3.1 is a simple generalisation of Mozannar and Sontag [48, Proposition 2], with the latter assuming $c = 0$. In Appendix F, we generalise Proposition 3.1 to the cascades of $K > 2$ models.

## 4 From Confidence-Based to Post-Hoc Deferral

Having presented the optimal deferral rule in Proposition 3.1, we now use it to explicate some failure modes for confidence-based deferral, which will be empirically demonstrated in §5.2.

### 4.1 When Does Confidence-Based Deferral Suffice?

Suppose as before we have probabilistic models $p^{(1)}, p^{(2)}$. Recall from Algorithm 1 that for constant $c^{(1)} > 0$, confidence-based deferral employs the rule

$$\hat{r}_{\text{conf}}(x) = 1\left[\max_{y'} p_{y'}^{(1)}(x) < c^{(1)}\right]. \tag{7}$$

Following §3.3, (7) may be regarded as a plug-in estimator for the "population confidence" rule $r_{\text{conf}}(x) \doteq 1\left[\eta_{h^{(1)}}(x) < c^{(1)}\right]$. Contrasting this to Proposition 3.1, we have the following:

**Lemma 4.1.** *Assume that for any $x, x' \in \mathcal{X}$, $\eta_{h^{(1)}}(x) \leq \eta_{h^{(1)}}(x')$ if and only if $\eta_{h^{(1)}}(x) - \eta_{h^{(2)}}(x) \leq \eta_{h^{(1)}}(x') - \eta_{h^{(2)}}(x')$ (i.e., $\eta_{h^{(1)}}(x)$ and $\eta_{h^{(1)}}(x) - \eta_{h^{(2)}}(x)$ produce the same ordering over instances $x \in \mathcal{X}$). Then, the deferral rule $r_{\text{conf}}$ produces the same deferral curve as the Bayes-optimal rule (3).*

Lemma 4.1 studies the agreement between the *deferral curves* of $r_{\text{conf}}$ and the Bayes-optimal solution, which eliminates the need for committing to a specific cost $c^{(1)}$. The lemma has an intuitive interpretation: population confidence-based deferral is optimal if and only if the *absolute* confidence in model 1's prediction agrees with the *relative* confidence is model 1 versus model 2's prediction.

Based on this, we now detail some cases where confidence-based deferral succeeds or fails.

**Success mode: expert $h^{(2)}$.** Lemma 4.1 has one immediate, intuitive consequence: *confidence-based deferral is optimal when the downstream model has a constant error probability*, i.e., $\eta_{h_2(x)}(x)$ is a constant for all $x \in \mathcal{X}$. This may happen, e.g., if that the labels are deterministic given the inputs, and the second classifier $h^{(2)}$ perfectly predicts them. Importantly, note that this is a sufficient (but *not* necessary) condition for the optimality of confidence-based deferral.

**Failure mode: specialist $h^{(2)}$.** As a converse to the above, one setting where confidence-based deferral may fail is when when the downstream model is a *specialist*, which performs well only on a

| Training label $z$ | Loss | Deferral rule | Method label | Comment |
|---|---|---|---|---|
| $1[y = h^{(2)}(x)] - 1[y = h^{(1)}(x)]$ | Mean-squared error | $g(x) > c$ | `Diff-01` | Regression |
| $p_y^{(2)}(x) - p_y^{(1)}(x)$ | Mean absolute error | $g(x) > c$ | `Diff-Prob` | Regression |
| $\max_{y'} p_{y'}^{(2)}(x)$ | Mean-squared error | $g(x) - \max_{y'} p_{y'}^{(1)}(x) > c$ | `MaxProb` | Regression |

Table 1: Candidate post-hoc estimators of the oracle rule in (2). We train a post-hoc model $g(x)$ on a training set $\{(x_i, z_i)\}_{i=1}^n$ so as to predict the label $z$. Here, $(x, y) \in \mathcal{X} \times \mathcal{Y}$ is a labeled example.

particular sub-group of the data (e.g., a subset of classes). Intuitively, confidence-based deferral may *erroneously forward samples where $h^{(2)}$ performs worse than $h^{(1)}$*.

Concretely, suppose there is a data sub-group $\mathcal{X}_{\text{good}} \subset \mathcal{X}$ where $h^{(2)}$ performs exceptionally well, i.e., $\eta_{h^{(2)}(x)} \approx 1$ when $x \in \mathcal{X}_{\text{good}}$. On the other hand, suppose $h^{(2)}$ does not perform well on $\mathcal{X}_{\text{bad}} \doteq \mathcal{X} \setminus \mathcal{X}_{\text{good}}$, i.e., $\eta_{h^{(2)}(x)} \approx 1/L$ when $x \in \mathcal{X}_{\text{bad}}$. Intuitively, while $\eta_{h^{(1)}(x)}$ may be relatively low for $x \in \mathcal{X}_{\text{bad}}$, it is strongly desirable to *not* defer such examples, as $h^{(2)}$ performs even worse than $h^{(1)}$; rather, it is preferable to identify and defer samples $x \in \mathcal{X}_{\text{good}}$.

**Failure mode: label noise**. Confidence-based deferral can fail when there are high levels of label noise. Intuitively, in such settings, confidence-based deferral may *wastefully forward samples where $h^{(2)}$ performs no better than $h^{(1)}$*. Concretely, suppose that instances $x \in \mathcal{X}_{\text{bad}} \subset \mathcal{X}$ may be mislabeled as one from a different, random class. For $x \in \mathcal{X}_{\text{bad}}$, regardless of how the two models $h^{(1)}, h^{(2)}$ perform, we have $\eta_{h^{(1)}(x)}(x), \eta_{h^{(2)}(x)}(x) = 1/L$ (i.e., the accuracy of classifying these instances is chance level in expectation). Since $\eta_{h^{(1)}(x)}(x)$ is low, confidence-based deferral will tend to defer such input instance $x$. However, this is a sub-optimal decision since model 2 is more computationally expensive, and expected to have the same chance-level performance.

**Failure mode: distribution shift**. Even when model $h^{(2)}$ is an expert model, an intuitive setting where confidence-based deferral can fail is if there is *distribution shift* between the train and test $\mathbb{P}(y \mid x)$ [56]. In such settings, even if $p^{(1)}$ produces reasonable estimates of the *training* class-probability, these may translate poorly to the test set. There are numerous examples of confidence degradation under such shifts, such as the presence of *out-of-distribution* samples [52, 28], and the presence of a *label skew* during training [79, 64]. We shall focus on the latter in the sequel.

### 4.2 Post-Hoc Estimates of the Deferral Rule

Having established that confidence-based deferral may be sub-optimal in certain settings, we now consider the viability of deferral rules that are *learned* in a post-hoc manner. Compared to confidence-based deferral, such rules aim to explicitly account for *both* the confidence of model 1 and 2, and thus avoid the failure cases identified above.

The key idea behind such post-hoc rules is to directly mimic the optimal deferral rule in (3). Recall that this optimal rule has a dependence on the output of $h^{(2)}$; unfortunately, querying $h^{(2)}$ defeats the entire purpose of cascades. Thus, our goal is to estimate (3) using only the outputs of $p^{(1)}$.

We summarise a number of post-hoc estimators in Table 1, which are directly motivated by the One-hot, Probability, and Relative Confidence Oracle respectively from §3.3. The first is to learn when model 1 is incorrect, and model 2 is correct. For example, given a validation set, suppose we construct samples $S_{\text{val}} \doteq \{(x_i, z_i^{(1)})\}$, where $z_i^{(1)} = 1[y = h^{(2)}(x_i)] - 1[y = h^{(1)}(x_i)]$. Then, we fit

$$\min_{g \colon \mathcal{X} \to \mathbb{R}} \frac{1}{|S_{\text{val}}|} \sum_{(x_i, z_i^{(1)}) \in S_{\text{val}}} \ell(z_i^{(1)}, g(x_i)),$$

where, e.g., $\ell$ is the square loss. The score $g(x)$ may be regarded as the confidence in deferring to model 2. Similarly, a second approach is to perform regression to predict $z_i^{(2)} = p_y^{(2)}(x_i) - p_y^{(1)}(x_i)$. The third approach is to directly estimate $z_i^{(3)} = \max_{y'} p_{y'}^{(2)}(x_i)$ using predictions of the first model.

As shall be seen in §5.2, such post-hoc rules can learn to avoid the failure cases for confidence-based deferral identified in the previous section. However, it is important to note that there are some conditions where such rules may not offer benefits over confidence-based deferral.

**Failure mode: Bayes** $p^{(2)}$. Suppose that the model $p^{(2)}$ exactly matches the Bayes-probabilities, i.e., $p_{y'}^{(2)}(x) = \mathbb{P}(y' \mid x)$. Then, estimating $\max_{y'} p_{y'}^{(2)}(x)$ is equivalent to estimating $\max_{y'} \mathbb{P}(y' \mid x)$. However, the goal of model 1 is *precisely* to estimate $\mathbb{P}(y \mid x)$. Thus, if $p^{(1)}$ is sufficiently accurate, in the absence of additional information (e.g., a fresh dataset), it is unlikely that one can obtain a better estimate of this probability than that provided by $p^{(1)}$ itself. This holds even if the $\mathbb{P}(y \mid x)$ is non-deterministic, and so the second model has non-trivial error.

**Failure mode: non-predictable** $p^{(2)}$ **error**. When the model $p^{(2)}$'s outputs are not strongly predictable, post-hoc deferral may devolve to regular confidence-based deferral. Formally, suppose we seek to predict $z \doteq \max_{y'} p_{y'}^{(2)}(x)$, e.g., as in MaxProb. A non-trivial predictor must achieve an average square error smaller than the variance of $z$, i.e., $\mathbb{E}[(z - \mathbb{E}[z])^2]$. If $z$ is however not strongly predictable, the estimate will be tantamount to simply using the constant $\mathbb{E}[z]$. This brings us back to the assumption of model 2 having a constant probability of error, i.e., confidence-based deferral.

### 4.3 Finite-Sample Analysis for Post-Hoc Deferral Rules

We now formally quantify the gap in performance between the learned post-hoc rule and the Bayes-optimal rule $r^* = 1[g^*(x) > c]$ in Proposition 3.1, where $g^*(x) = \eta_{h^{(2)}(x)}(x) - \eta_{h^{(1)}(x)}(x)$. We will consider the case where the validation sample $S_{\mathrm{val}}$ is constructed with labels $z_i^{(1)}$. We pick a scorer $\hat{g}$ that minimises the average squared loss $\frac{1}{|S_{\mathrm{val}}|} \sum_{(x_i,z_i) \in S_{\mathrm{val}}} (z_i - g(x_i))^2$ over a hypothesis class $\mathcal{G}$. We then construct a deferral rule $\hat{r}(x) = 1[\hat{g}(x) > c]$.

**Lemma 4.2.** *Let $\mathcal{N}(\mathcal{G}, \epsilon)$ denote the covering number of $\mathcal{G}$ with the $\infty$-norm. Suppose for any $g \in \mathcal{G}$, $(z - g(x))^2 \leq B, \forall(x, z)$. Furthermore, let $\tilde{g}$ denote the minimizer of the population squared loss $\mathbb{E}\left[(z - g(x))^2\right]$ over $\mathcal{G}$, where $z = 1[y = h^{(2)}(x)] - 1[y = h^{(1)}(x)]$. Then for any $\delta \in (0, 1)$, with probability at least $1 - \delta$ over draw of $S_{\mathrm{val}}$, the excess risk for $\hat{r}$ is bounded by*

$$R(\hat{r}; h^{(1)}, h^{(2)}) - R(r^*; h^{(1)}, h^{(2)})$$

$$\leq 2 \cdot \left( \underbrace{\mathbb{E}_x\left[(\tilde{g}(x) - g^*(x))^2\right]}_{\text{Approximation error}} + \underbrace{4 \cdot \inf_{\epsilon > 0}\left\{ B\sqrt{\frac{2 \cdot \log \mathcal{N}(\mathcal{G}, \epsilon)}{|S_{\mathrm{val}}|}}\right\} + \mathcal{O}\left(\sqrt{\frac{\log(1/\delta)}{|S_{\mathrm{val}}|}}\right)}_{\text{Estimation error}} \right)^{1/2}.$$

We provide the proof in §A.4. The first term on the right-hand side (RHS) is an irreducible approximation error, quantifying the distance between the best possible model in the class to the oracle scoring function. The second and the third terms on RHS quantify total estimation error.

### 4.4 Relation to Existing Work

As noted in §3.4, learning a deferral rule for a two-model cascade is closely related to existing literature in learning to defer to an expert. This in turn is a generalisation of the classical literature on *learning to reject* [27, 7], which refers to classification settings where one is allowed to abstain from predicting on certain inputs. The population risk here is a special case of (2), where $h^{(2)}(x)$ is assumed to perfectly predict $y$. The resulting Bayes-optimal classifier is known as Chow's rule [11, 60], and exactly coincides with the deferral rule in Lemma 4.1. Plug-in estimates of this rule are thus analogous to confidence-based deferral, and have been shown to be similarly effective [53].

In settings where one is allowed to modify the training of $h^{(1)}$, it is possible to construct losses that jointly optimise for both $h^{(1)}$ and $r$ [1, 12, 60, 73, 8, 21, 36]. While effective, these are not applicable in our setting involving pre-trained, black-box classifiers. Other variants of post-hoc methods have been considered in Narasimhan et al. [49], and implicitly in Trapeznikov and Saligrama [74]; however, here we more carefully study the different possible ways of constructing these methods, and highlight when they may fail to improve over confidence-based deferral.

## 5 Experimental Illustration

In this section, we provide empirical evidence to support our analysis in §4.1 by considering the three failure modes in which confidence-based deferral underperforms. For each of these settings, we

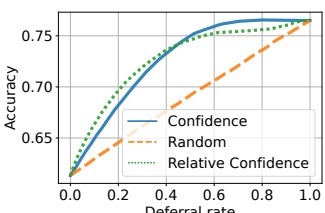 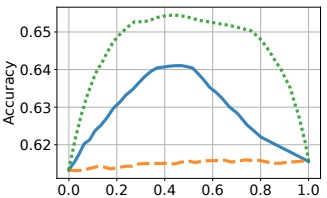 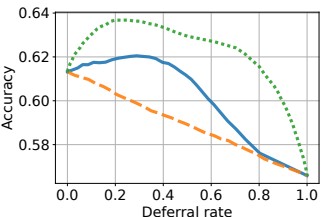

(a) Standard ImageNet
(100% non-dog training images)

(b) 4% non-dog training images

(c) 2% non-dog training images

Figure 1: Test accuracy vs deferral rate of plug-in estimates (§3.3) for the oracle rule. Here, $h^{(1)}$ is a MobileNet V2 trained on all ImageNet classes, and $h^{(2)}$ is a dog specialist trained on all images in the dog synset plus a fraction of non-dog training examples, which we vary. As the fraction decreases, $h^{(2)}$ specialises in classifying different types of dogs. By considering the confidence of $h^{(2)}$ (Relative Confidence), one gains accuracy by selectively deferring only dog images.

compute *deferral curves* that plot the classification accuracy versus the fraction of samples deferred to the second model (which implicitly measures the overall compute cost). In line with our analysis in §4.1, post-hoc deferral rules offer better accuracy-cost trade-offs in these settings.

## 5.1 Confidence-Based versus Oracle Deferral

We begin by illustrating the benefit of considering confidence of the second model when constructing a deferral rule. In this experiment, $h^{(1)}$ is a generalist (i.e., trained on all ImageNet classes), and $h^{(2)}$ is a dog specialist trained on all images in the dog synset, plus a fraction of non-dog training examples, which we vary. There are 119 classes in the dog synset. We use MobileNet V2 [65] as $h^{(1)}$, and a larger EfficientNet B0 [71] as $h^{(2)}$. For hyperparameter details, see Appendix C.

Figure 1 shows the accuracy of confidence-based deferral (Confidence) and Relative Confidence (Equation (6)) on the standard ImageNet test set as a function of the deferral rate. We realise different deferral rates by varying the value of the deferral threshold $c$. In Figure 1a, the fraction of non-dog training images is 100% i.e., model 2 is also a generalist trained on all images. In this case, we observe that Relative Confidence offers little gains over Confidence.

However, in Figure 1b and Figure 1c, as the fraction of non-dog training images decreases, the non-uniformity of $h^{(2)}$'s error probabilities increases i.e., $h^{(2)}$ starts to specialise to dog images. In line with our analysis in §4.1, confidence-based deferral underperforms when model 2's error probability is highly non-uniform. That is, being oblivious to the fact that $h^{(2)}$ specialises in dog images, confidence-based deferral may erroneously defer non-dog images to it. By contrast, accounting for model 2's confidence, as done by Relative Confidence, shows significant gains.

## 5.2 Confidence-Based versus Post-Hoc Deferral

From §5.1, one may construct better deferral rules by querying model 2 for its confidence. Practically, however, querying model 2 at inference time defeats the entire purpose of cascades. To that end, we now compare confidence-based deferral and the post-hoc estimators (Table 1), which do *not* need to invoke model 2 at inference time. We consider each of the settings from §4.1, and demonstrate that post-hoc deferral can significantly outperform confidence-based deferral. We present more experimental results in Appendix E, where we illustrate post-hoc deferral rules for $K > 2$ models.

**Post hoc model training.** For a post-hoc approach to be practical, the overhead from invoking a post-hoc model must be small relative to the costs of $h^{(1)}$ and $h^{(2)}$. To this end, in all of the following experiments, the post-hoc model $g\colon \mathcal{X} \to \mathbb{R}$ is based on a lightweight, three-layer Multi-Layer Perceptron (MLP) that takes as input the probability outputs from model 1. That is, $g(x) = \mathrm{MLP}(p^{(1)}(x))$ where $p^{(1)}(x) \in \Delta_L$ denotes all probability outputs from model 1. Learning $g$ amounts to learning the MLP as the two base models are fixed. We train $g$ on a held-out validation set. For full technical details of the post-hoc model architecture and training, see Appendix C. We use the objectives described in Table 1 to train $g$.

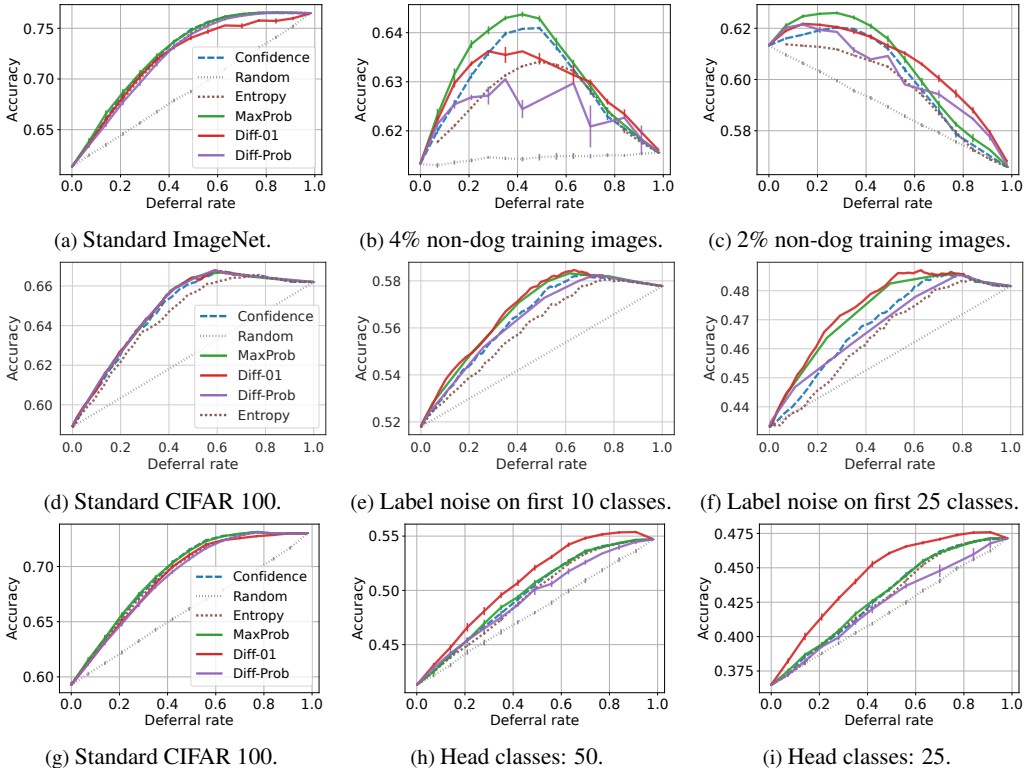


(a) Standard ImageNet.     (b) 4% non-dog training images.     (c) 2% non-dog training images.



(d) Standard CIFAR 100.     (e) Label noise on first 10 classes.     (f) Label noise on first 25 classes.



(g) Standard CIFAR 100.     (h) Head classes: 50.     (i) Head classes: 25.


Figure 2: Test accuracy vs deferral rate of the post-hoc approaches in Table 1 under the three settings described in §4.1: 1) specialist (row 1), 2) label noise (row 2), and 3) distribution shift. **Row 1**: As the fraction of non-dog training images decreases, model 2 becomes a dog specialist model. Increase in the non-uniformity in its error probabilities allows post-hoc approaches to learn to only defer dog images. **Row 2**: As label noise increases, the difference in the probability of correct prediction under each model becomes zero (i.e., probability tends to chance level). Thus, it is sub-optimal to defer affected inputs since model 2's correctness is also at chance level. Being oblivious to model 2, confidence-based deferral underperforms. For full details, see §5.2. **Row 3**: As the skewness of the label distribution increases, so does the difference in the probability of correct prediction under each model (recall the optimal rule in Proposition 3.1), and it becomes necessary to account for model 2's probability of correct prediction when deferring. Hence, confidence-based deferral underperforms.

**Specialist setting**. We start with the same ImageNet-Dog specialist setting used in §5.1. This time, we compare six methods: Confidence, Random, MaxProb, Diff-01, Diff-Prob, and Entropy. Random is a baseline approach that defers to either model 1 or model 2 at random; MaxProb, Diff-01, and Diff-Prob are the post-hoc rules described in Table 1; Entropy defers based on the thresholding the entropy of $p^{(1)}$ [31, 36], as opposed to the maximum probability.

Results for this setting are presented in Figure 2 (first row). We see that there are gains from post-hoc deferral, especially in the low deferral regime: it can accurately determine whether the second model is likely to make a mistake. Aligning with our analysis, for the generalist setting (Figure 2a), confidence-based deferral is highly competitive, since $h^{(2)}$ gives a consistent estimate of $\mathbb{P}(y \mid x)$.

**Label noise setting**. In this setting, we look at a problem with label noise. We consider CIFAR 100 dataset where training examples from pre-chosen $L_{\text{noise}} \in \{0, 10, 25\}$ classes are assigned a uniformly drawn label. The case of $L_{\text{noise}} = 0$ corresponds to the standard CIFAR 100 problem. We set $h^{(1)}$ to be CIFAR ResNet 8 and set $h^{(2)}$ to be CIFAR ResNet 14, and train both models on the noisy data. The results are shown in Figure 2 (second row). It is evident that when there is label noise, post-hoc approaches yield higher accuracy than confidence-based on a large range of deferral rates, aligning with our analysis in §4.1. Intuitively, confidence-based deferral tends to forward noisy samples to $h^{(2)}$, which performs equally poorly, thus leading to a waste of deferral budget. By contrast, post-hoc rules can learn to "give up" on samples with extremely low model 1 confidence.

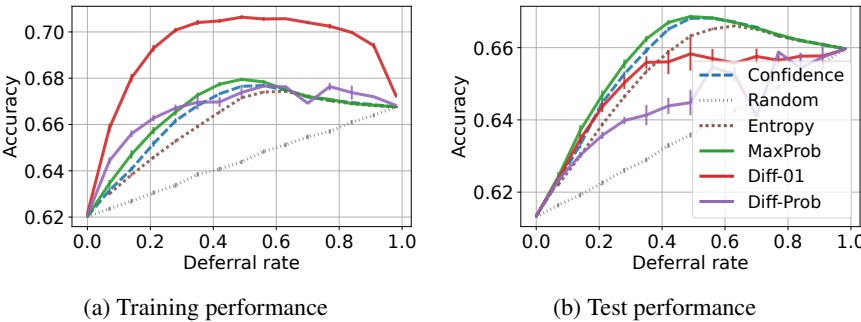

(a) Training performance      (b) Test performance

Figure 3: Training and test accuracy of post-hoc approaches in the ImageNet-Dog specialist setting. Model 2 (EfficientNet B0) is trained with all dog images and 8% of non-dog images. Observe that a post-hoc model (i.e., `Diff-01`) can severely overfit to the training set and fail to generalise.

**Distribution shift setting**. To simulate distribution shift, we consider a long-tailed version of CIFAR 100 [40] where there are $h \in \{100, 50, 25\}$ head classes, and $100 - h$ tail classes. Each head class has 500 training images, and each tail class has 50 training images. The standard CIFAR 100 dataset corresponds to $h = 100$. Both models $h^{(1)}$ (CIFAR ResNet 8) and $h^{(2)}$ (CIFAR ResNet 56) are trained on these long-tailed datasets. At test time, all methods are evaluated on the standard CIFAR 100 balanced test set, resulting in a label distribution shift.

We present our results in Figure 2 (third row). In Figure 2g, there is no distribution shift. As in the case of the specialist setting, there is little to no gain from post-hoc approaches in this case since both models are sufficiently accurate. As $h$ decreases from 100 to 50 (Figure 2h) and 25 (Figure 2i), there is more distribution shift at test time, and post-hoc approaches (notably `Diff-01`) show clearer gains. To elaborate, the two base models are of different sizes and respond to the distribution shift differently, with CIFAR ResNet 56 being able to better handle tail classes overall. `Diff-01` is able to identify the superior performance of $h^{(2)}$ and defer input instances from tail classes.

### 5.3 On the Generalisation of Post-Hoc Estimators

Despite the benefits of post-hoc approaches as demonstrated earlier, care must be taken in controlling the capacity of the post-hoc models. We consider the same ImageNet-Dog specialist setting as in the top row of Figure 2. Here, model 2 is trained on all dog images, and a large fraction of non-dog images (8%). Since model 2 has access to a non-trivial fraction of non-dog images, the difference in the probability of correct prediction of the two models is less predictable. We report deferral curves on both training and test splits in Figure 3. Indeed, we observe that the post-hoc method `Diff-01` can overfit, and fail to generalise. Note that this is despite using a feedforward network with two hidden layers of only 64 and 16 units (see Appendix C for details on hyperparameters) to control the capacity of the post-hoc model. Thoroughly investigating approaches to increase generalisation of post-hoc models will be an interesting topic for future study.

## 6 Conclusion and Future Work

The Bayes-optimal deferral rule we present suggests that key to optimally defer is to identify when the first model is wrong and the second is right. Based on this result, we then study a number of estimators (Table 1) to construct trainable post hoc deferral rules, and show that they can improve upon the commonly used confidence-based deferral.

While we have identified conditions under which confidence-based deferral underperforms (e.g., specialist setting, label noise), these are not exhaustive. An interesting direction for future work is to design post-hoc deferral schemes attuned for settings involving other forms of distribution shift, such as the presence of out-of-distribution samples. It is also of interest to study the efficacy of more refined confidence measures, such as those based on conformal prediction [69]. Finally, while our results have focussed on image classification settings, it would be of interest to study analogous trends for natural language processing models.

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
