# When Does Confidence-Based Cascade Deferral Suffice?

## Appendix

## Table of Contents

## A  Proofs

### A.1  Proof of Proposition 3.1

*Proof of Proposition 3.1.* The risk in (2) can be written as

$R(r; h^{(1)}, h^{(2)})$

$= \mathbb{P}(y \neq h^{(1)}(x), r(x) = 0) + \mathbb{P}(y \neq h^{(2)}(x), r(x) = 1) + c \cdot \mathbb{P}(r(x) = 1)$

$= \mathbb{E}\left[ 1[y \neq h^{(1)}(x)] \cdot 1[r(x) = 0] + 1[y \neq h^{(2)}(x)] \cdot 1[r(x) = 1] + c1[r(x) = 1] \right]$

$= \mathbb{E}\left[ 1[y \neq h^{(1)}(x)] \cdot (1 - 1[r(x) = 1]) + 1[y \neq h^{(2)}(x)] \cdot 1[r(x) = 1] + c1[r(x) = 1] \right]$

$= \mathbb{P}(y \neq h^{(1)}(x)) + \mathbb{E}\left[ -1[y \neq h^{(1)}(x)] \cdot 1[r(x) = 1] + 1[y \neq h^{(2)}(x)] \cdot 1[r(x) = 1] + c1[r(x) = 1] \right]$

$= \mathbb{P}(y \neq h^{(1)}(x)) + \mathbb{E}_x 1[r(x) = 1] \mathbb{E}_{y|x}\left[ 1[y \neq h^{(2)}(x)] - 1[y \neq h^{(1)}(x)] + c \right]$

$= \mathbb{P}(y \neq h^{(1)}(x)) + \mathbb{E}_x 1[r(x) = 1] \mathbb{E}_{y|x}\left[ 1[y = h^{(1)}(x)] - 1[y = h^{(2)}(x)] + c \right]$

$= \mathbb{P}(y \neq h^{(1)}(x)) + \mathbb{E}_x 1[r(x) = 1] \cdot \left[ \eta_{h^{(1)}(x)}(x) - \eta_{h^{(2)}(x)}(x) + c \right],$

where we define $\eta_{y'}(x) \doteq \mathbb{P}(y = y' \mid x)$. Thus, it is optimal to defer when

$$r(x) = 1 \iff \eta_{h^{(1)}(x)}(x) - \eta_{h^{(2)}(x)}(x) + c < 0$$

$$\iff \eta_{h^{(2)}(x)}(x) - \eta_{h^{(1)}(x)}(x) > c.$$

$\square$

## A.2 Proof of Corollary 3.2

*Proof of Corollary 3.2.* For fixed $h^{(1)}, h^{(2)}$, we have already computed the Bayes-optimal rejector $r^*$. Let $\alpha(x) \doteq \eta_{h^{(1)}(x)}(x) - \eta_{h^{(2)}(x)}(x) + c$. Plugging $r^*$ into the risk results in Bayes-risk

$$R(r^*; h^{(1)}, h^{(2)}) = \mathbb{P}(y \neq h^{(1)}(x)) + \mathbb{E}_x \mathbf{1}\left[r^*(x) = 1\right] \cdot \alpha(x).$$

The excess risk for an arbitrary $r$ is thus

$$R(r; h^{(1)}, h^{(2)}) - R(r^*; h^{(1)}, h^{(2)})$$
$$= \mathbb{E}_x \left[\mathbf{1}\left[r(x) = 1\right] - \mathbf{1}\left[\alpha(x) < 0\right]\right] \cdot \alpha(x).$$

$\square$

## A.3 Proof of Lemma 4.1

We start with Lemma A.1 which will help prove Lemma 4.1.

**Lemma A.1.** *Given two base classifiers $h^{(1)}, h^{(2)} : \mathcal{X} \to [L]$, two deferral rules $r_1, r_2 : \mathcal{X} \to \{0, 1\}$ yield the same accuracy for the cascade if and only if $\mathbb{E}(r_1(x)\beta(x)) = \mathbb{E}(r_2(x)\beta(x))$, where $\beta(x) \doteq \eta_{h^{(2)}(x)}(x) - \eta_{h^{(1)}(x)}(x)$.*

*Proof.* For a deferral rule $r$, by definition, the accuracy is given by

$$A(r) = \mathbb{E}_{(x,y)} \left[\mathbf{1}[h^{(1)}(x) = y] \cdot (1 - r(x)) + \mathbf{1}[h^{(2)}(x) = y] \cdot r(x)\right]$$
$$= \mathbb{P}(h^{(1)}(x) = y) + \mathbb{E}\left[r(x) \cdot \left(\mathbf{1}[h^{(2)}(x) = y] - \mathbf{1}[h^{(1)}(x) = y]\right)\right]$$
$$= \mathbb{P}(h^{(1)}(x) = y) + \mathbb{E}_x r(x) \mathbb{E}_{y|x}\left[\mathbf{1}[h^{(2)}(x) = y] - \mathbf{1}[h^{(1)}(x) = y]\right]$$
$$= \mathbb{P}(h^{(1)}(x) = y) + \mathbb{E}_x r(x) \cdot \left(\eta_{h^{(2)}(x)}(x) - \eta_{h^{(1)}(x)}(x)\right).$$

Given two deferral rules $r_1, r_2$, $A(r_1) = A(r_2) \iff \mathbb{E}(r_1(x)\beta(x)) = \mathbb{E}(r_2(x)\beta(x))$ where $\beta(x) \doteq \eta_{h^{(2)}(x)}(x) - \eta_{h^{(1)}(x)}(x)$. $\square$

We are ready to prove Lemma 4.1.

*Proof of Lemma 4.1.* Recall the confidence-based deferral rule and the Bayes-optimal rule are respectively

$$r_{\text{conf}}(x) = \mathbf{1}\left[\eta_{h^{(1)}(x)}(x) < c'\right],$$
$$r^*(x) = \mathbf{1}\left[\eta_{h^{(1)}(x)}(x) - \eta_{h^{(2)}(x)}(x) < c\right].$$

For brevity, we use $\eta_i(x)$ and $\eta_{h^{(i)}(x)}(x)$ interchangeably. Define real-valued random variables $z \doteq \eta_1(x)$, and $z^* \doteq \eta_1(x) - \eta_2(x)$ where $x \sim \mathbb{P}_x$. Let $\rho(c') \doteq \mathbb{E}(\mathbf{1}[\eta_1(x) < c']) = \mathbb{P}(\eta_1(x) < c')$ be the deferral rate of the confidence-based deferral rule with threshold $c'$. Similarly, define $\rho^*(c) \doteq \mathbb{E}(\mathbf{1}[\eta_1(x) - \eta_2(x) < c])$. Let $\gamma_\alpha, \gamma_\alpha^*$ be the $\alpha$-quantile of the distributions of $z$ and $z^*$, respectively. By definition, $\rho(\gamma_\alpha) = \rho^*(\gamma_\alpha^*) = \alpha$.

Let $A(r, c')$ denote the accuracy of the cascade with the deferral rule $r$ and the deferral threshold $c'$. Formally, the deferral curve of $r_{\text{conf}}$ is the set of deferral rate-accuracy tuples $\{(\rho(c'), A(r_{\text{conf}}, c')) \mid c' \in [0, 1]\}$. The same deferral curve may be generated with $\{(\alpha, A(r_{\text{conf}}, \gamma_\alpha)) \mid \alpha \in [0, 1]\}$. Similarly, for the Bayes-optimal rule, the deferral curve is defined as $\{(\alpha, A(r^*, \gamma_\alpha^*) \mid \alpha \in [0, 1]\}$.

To show that the two deferral rules produce the same deferral curve, we show that $A(r_{\text{conf}}, \gamma_\alpha) = A(r^*, \gamma_\alpha^*)$ for any $\alpha \in [0, 1]$. By Lemma A.1, this is equivalent to showing that

$$\mathbb{E}(\mathbf{1}[\eta_1(x) < \gamma_\alpha] \cdot \beta(x)) = \mathbb{E}(\mathbf{1}[\eta_1(x) - \eta_2(x) < \gamma_\alpha^*] \cdot \beta(x)), \tag{8}$$

where $\beta(x) \doteq \eta_2(x) - \eta_1(x)$.

Suppose that $\eta_1(x)$ and $\eta_1(x) - \eta_2(x)$ produce the same ordering over instances $x \in \mathcal{X}$. This means that for any $x \in \mathcal{X}$ and $\alpha \in [0, 1]$, $\eta_1(x) < \gamma_\alpha \iff \eta_1(x) - \eta_2(x) < \gamma_\alpha^*$. Thus, (8) holds. $\square$

### A.4 Proof of Lemma 4.2

We provide an excess risk bound in Lemma A.2 and generalization bound in Lemma A.3. The proof of Lemma 4.2 follows from combining the two results.

**Excess risk bound.** The excess risk for the learned deferral rule can be bounded as follows:

**Lemma A.2.** *The excess risk for $\hat{r}$ is bounded by:*
$$R(\hat{r}; h^{(1)}, h^{(2)}) - R(r^*; h^{(1)}, h^{(2)}) \leq 2\sqrt{\mathbb{E}_x\left[(\hat{g}(x) - g^*(x))^2\right]}.$$

*Proof.* We first expand the risk for deferral rule $\hat{r}$:
$$R(\hat{r}; h^{(1)}, h^{(2)}) = \mathbb{E}\left[1[y \neq h^{(1)}(x), \hat{g}(x) \leq c] + 1[y \neq h^{(2)}(x), \hat{g}(x) > c] + c \cdot 1[\hat{g}(x) > c]\right]$$
$$= \mathbb{E}_x\left[(1 - \eta_{h^{(1)}(x)}(x)) \cdot 1[\hat{g}(x) \leq c] + (1 - \eta_{h^{(2)}(x)}(x) + c) \cdot 1[\hat{g}(x) > c]\right]$$
$$= \mathbb{E}_x\left[(\eta_{h^{(1)}(x)}(x) - \eta_{h^{(2)}(x)}(x) + c) \cdot 1[\hat{g}(x) > c]\right] + \mathbb{E}_x\left[1 - \eta_{h^{(1)}(x)}(x)\right]$$
$$= \mathbb{E}_x\left[(-g^*(x) + c) \cdot 1[\hat{g}(x) > c]\right] + \mathbb{E}_x\left[1 - \eta_{h^{(1)}(x)}(x)\right].$$

Per Corollary 3.2, the excess risk for $\hat{r}$ can then be written as:
$$R(\hat{r}; h^{(1)}, h^{(2)}) - R(r^*; h^{(1)}, h^{(2)})$$
$$= \mathbb{E}_x\left[(-g^*(x) + c) \cdot 1[\hat{g}(x) > c]\right] - \mathbb{E}_x\left[(-g^*(x) + c) \cdot 1[g^*(x) > c]\right]$$
$$= \mathbb{E}_x\left[(-g^*(x) + c) \cdot 1[\hat{g}(x) > c]\right] - \mathbb{E}_x\left[(-\hat{g}(x) + c) \cdot 1[g^*(x) > c]\right]$$
$$\quad + \mathbb{E}_x\left[(-\hat{g}(x) + c) \cdot 1[g^*(x) > c]\right] - \mathbb{E}_x\left[(-g^*(x) + c) \cdot 1[g^*(x) > c]\right].$$

We next bound the second term on the right-hand side. Note that $(-\hat{g}(x) + c) \cdot 1[g'(x) > c]$ is minimized when $g'(x) = \hat{g}(x)$. Therefore $(-\hat{g}(x) + c) \cdot 1[g^*(x) > c] \geq (-\hat{g}(x) + c) \cdot 1[\hat{g}(x) > c]$. Plugging this into the above inequality gives us:
$$R(\hat{r}; h^{(1)}, h^{(2)}) - R(r^*; h^{(1)}, h^{(2)})$$
$$\leq \mathbb{E}_x\left[(-g^*(x) + c) \cdot 1[\hat{g}(x) > c]\right] - \mathbb{E}_x\left[(-\hat{g}(x) + c) \cdot 1[\hat{g}(x) > c]\right]$$
$$\quad + \mathbb{E}_x\left[(-\hat{g}(x) + c) \cdot 1[g^*(x) > c]\right] - \mathbb{E}_x\left[(-g^*(x) + c) \cdot 1[g^*(x) > c]\right]$$
$$= \mathbb{E}_x\left[(\hat{g}(x) - g^*(x)) \cdot 1[\hat{g}(x) > c]\right] + \mathbb{E}_x\left[(g^*(x) - \hat{g}(x)) \cdot 1[g^*(x) > c]\right]$$
$$\leq \mathbb{E}_x\left[|\hat{g}(x) - g^*(x)| \cdot (1[\hat{g}(x) > c] + 1[g^*(x) > c])\right]$$
$$\leq 2 \cdot \mathbb{E}_x\left[|\hat{g}(x) - g^*(x)|\right]$$
$$\leq 2 \cdot \mathbb{E}_x\left[\sqrt{(\hat{g}(x) - g^*(x))^2}\right]$$
$$\leq 2 \cdot \sqrt{\mathbb{E}_x\left[(\hat{g}(x) - g^*(x))^2\right]},$$
where the second-last step follows from Jensen's inequality and the fact that $\sqrt{\cdot}$ is concave. $\square$

**Finite-sample bound.** Next, we bound the right-hand side of the excess risk bound in Lemma A.2 with a finite-sample bound. For this, define the population squared loss for a scorer $g$ by:
$$R_{\text{sq}}(g) = \mathbb{E}\left[(z - g(x))^2\right],$$
where $z = 1[y = h^{(2)}(x)] - 1[y = h^{(1)}(x)]$. It is straight-forward to see that the minimiser of this loss over all measurable functions $g : \mathcal{X} \to \mathbb{R}$ is $\mathbb{E}[z|x] = \eta_{h^{(2)}(x)}(x) - \eta_{h^{(1)}(x)}(x) = g^*(x)$. Let $\hat{g}$ be minimizer of the average squared loss $\frac{1}{|S_{\text{val}}|}\sum_{(x_i, z_i) \in S_{\text{val}}}(z_i - g(x_i))^2$ over hypothesis class $\mathcal{G}$, where the labels in $S_{\text{val}}$ are given by $z_i = 1[y_i = h^{(2)}(x_i)] - 1[y_i = h^{(1)}(x_i)]$.

**Lemma A.3.** *Let $\mathcal{N}(\mathcal{G}, \epsilon)$ denote the covering number for $\mathcal{G}$ with the $\infty$-norm. Suppose for any $g \in \mathcal{G}$, the squared loss $(z - g(x))^2 \leq B, \forall (x, z)$. Furthermore, let $\tilde{g}$ denote the minimizer of the population squared loss $\mathbb{E}\left[(z - g(x))^2\right]$ over $\mathcal{G}$, where $z = 1[y = h^{(2)}(x)] - 1[y = h^{(1)}(x)]$. Then for any $\delta \in (0, 1)$, with probability at least $1 - \delta$ over draw of $S_{\text{val}}$,*
$$\mathbb{E}_x\left[(\hat{g}(x) - g^*(x))^2\right]$$
$$\leq \underbrace{\mathbb{E}_x\left[(\tilde{g}(x) - g^*(x))^2\right]}_{\textit{Approximation error}} + \underbrace{4 \cdot \inf_{\epsilon > 0}\left\{B\sqrt{\frac{2 \cdot \log \mathcal{N}(\mathcal{G}, \epsilon)}{|S_{\text{val}}|}}\right\} + \mathcal{O}\left(\sqrt{\frac{\log(1/\delta)}{|S_{\text{val}}|}}\right)}_{\textit{Estimation error}}.$$

*Proof.* We have from standard uniform convergence arguments (Shalev-Shwartz & Ben-David, 2014), the following generalization bound for the squared loss: with probability at least $1 - \delta$ over draw of $S_{\text{val}}$, for any $g \in \mathcal{G}$:

$$\left| R_{\text{sq}}(g) - \hat{R}_{\text{sq}}(g) \right| \leq 2 \cdot \inf_{\epsilon > 0} \left\{ \epsilon + B \sqrt{\frac{2 \cdot \log \mathcal{N}(\mathcal{G}, \epsilon)}{|S_{\text{val}}|}} \right\} + \mathcal{O}\left( \sqrt{\frac{\log(1/\delta)}{|S_{\text{val}}|}} \right). \tag{9}$$

Expanding the population squared loss for a scorer $g$, we have:

$$
\begin{aligned}
R_{\text{sq}}(g) &= \mathbb{E}_x \left[ \mathbb{E}_{z|x} \left[ (z - g(x))^2 \right] \right] \\
&= \mathbb{E}_x \left[ \mathbb{E}_{z|x} \left[ z^2 \right] - 2 \cdot \mathbb{E}_{z|x} \left[ z \right] \cdot g(x) + g(x)^2 \right] \\
&= \mathbb{E}_x \left[ \mathbb{E}_{z|x} \left[ z \right] - 2 \cdot \mathbb{E}_{z|x} \left[ z \right] \cdot g(x) + g(x)^2 \right] + \mathbb{E} \left[ z^2 - z \right] \\
&= \mathbb{E}_x \left[ (\mathbb{E}_{z|x} \left[ z \right] - g(x))^2 \right] + \mathbb{E} \left[ z^2 - z \right] \\
&= \mathbb{E}_x \left[ (g^*(x) - g(x))^2 \right] + \mathbb{E} \left[ z^2 - z \right].
\end{aligned}
$$

Notice that the second term is independent of $g$. The excess squared loss for any scorer $g$ can then be written as:

$$R_{\text{sq}}(g) - R_{\text{sq}}(g^*) = \mathbb{E}_x \left[ (g(x) - g^*(x))^2 \right]. \tag{10}$$

It then follows that:

$$
\begin{aligned}
\mathbb{E}_x \left[ (\hat{g}(x) - g^*(x))^2 \right] &= R_{\text{sq}}(\hat{g}) - R_{\text{sq}}(g^*) \\
&= R_{\text{sq}}(\hat{g}) - R_{\text{sq}}(\tilde{g}) + R_{\text{sq}}(\tilde{g}) - R_{\text{sq}}(g^*) \\
&\overset{(a)}{=} R_{\text{sq}}(\hat{g}) - R_{\text{sq}}(\tilde{g}) + \mathbb{E}_x \left[ (\tilde{g}(x) - g^*(x))^2 \right] \\
&= R_{\text{sq}}(\hat{g}) - \hat{R}_{\text{sq}}(\tilde{g}) + \hat{R}_{\text{sq}}(\tilde{g}) - R_{\text{sq}}(\tilde{g}) + \mathbb{E}_x \left[ (\tilde{g}(x) - g^*(x))^2 \right] \\
&\overset{(b)}{\leq} R_{\text{sq}}(\hat{g}) - \hat{R}_{\text{sq}}(\hat{g}) + \hat{R}_{\text{sq}}(\tilde{g}) - R_{\text{sq}}(\tilde{g}) + \mathbb{E}_x \left[ (\tilde{g}(x) - g^*(x))^2 \right] \\
&\leq |R_{\text{sq}}(\hat{g}) - \hat{R}_{\text{sq}}(\hat{g})| + |\hat{R}_{\text{sq}}(\tilde{g}) - R_{\text{sq}}(\tilde{g})| + \mathbb{E}_x \left[ (\tilde{g}(x) - g^*(x))^2 \right] \\
&\leq 2 \cdot \sup_{g \in \mathcal{G}} |R_{\text{sq}}(g) - \hat{R}_{\text{sq}}(g)| + \mathbb{E}_x \left[ (\tilde{g}(x) - g^*(x))^2 \right],
\end{aligned}
$$

where step $(a)$ uses (10); step $(b)$ uses the fact that $\hat{g}$ is the minimizer of $\hat{R}_{\text{sq}}$ over $\mathcal{G}$. Upper bounding the right-hand side using the generalization bound in (9) completes the proof. $\square$

## B  Amount of Compute Used for Experiments

There are two types of models involved in all experiments: (1) base classifiers (i.e., $h^{(1)}$ and $h^{(2)}$), and (2) post-hoc model. For post-hoc model training and evaluation, we use one Nvidia V100 GPU. As discussed in Appendix C.2, our post-hoc model is only a small MLP model with only two hidden layers. In each experiment reported in the main text, training one post-hoc model for 20 epochs only takes a few minutes. For training and evaluating a post-hoc model, a GPU is not needed.

For training of base classifiers, the amount of compute varies depending on the dataset and model architecture. This is summarized in the following table. In the following table, a GPU always refers to an Nvidia V100 GPU, and a TPU always refers to a Google Cloud TPU v3.[1]

| Dataset | Model | Devices | Approximate Training Time |
|---|---|---|---|
| CIFAR 100 | CIFAR ResNet 8 | $8\times$ GPUs | 12m (batch size: 1024, 256 epochs) |
| CIFAR 100 | CIFAR ResNet 14 | $8\times$ GPUs | 20m (batch size: 1024, 256 epochs) |
| CIFAR 100 | CIFAR ResNet 56 | $8\times$ GPUs | 20m (batch size: 1024, 256 epochs) |
| ImageNet | MobileNet V2 | $8\times$ TPUs | 7h (batch size: 64, 90 epochs) |
| ImageNet | EfficientNet B0 | $8\times$ TPUs | 4h (batch size: 1024, 90 epochs) |

[1]Google Cloud TPU v3: https://cloud.google.com/tpu/docs/system-architecture-tpu-vm.

# C Experimental Setup: Hyper-parameters

## C.1 Training of Models in a Cascade

We describe hyperparameters we used for training all base models (i.e., $h^{(1)}$ and $h^{(2)}$). In the following table, BS denotes batch size, and schedule refers to learning rate schedule.

| Dataset | Model | LR | Schedule | Epochs | BS |
|---|---|---|---|---|---|
| Standard | MobileNet V2 | 0.05 | anneal | 90 | 64 |
| ImageNet | EfficientNet B0 | 0.1 | cosine | 90 | 1024 |
| ImageNet Dog (specialist) | EfficientNet B0 | 0.1 | cosine | 90 | 512 |
| CIFAR 100 | CIFAR ResNet 8 | 1.0 | anneal | 256 | 1024 |
| | CIFAR ResNet 14 | 1.0 | anneal | 256 | 1024 |
| | CIFAR ResNet 56 | 1.0 | anneal | 256 | 1024 |

We use SGD with momentum as the optimisation algorithm for all models. For annealing schedule, the specified learning rate (LR) is the initial rate, which is then decayed by a factor of ten after each epoch in a specified list. For CIFAR100, these epochs are 15, 96, 192 and 224. For ImageNet, these epochs are 5, 30, 60, and 80. In the above table, CIFAR 100 includes the standard CIFAR 100 classification problem, CIFAR 100 with label noise, and CIFAR 100 with distribution shift as discussed in Figure 2.

## C.2 Training of Post-hoc Deferral Rules

All post-hoc models $g \colon \mathcal{X} \to \mathbb{R}$ we consider are based on a lightweight Multi-Layer Perceptron (MLP) that takes as input a small set of features constructed from probability outputs from model 1. More precisely, let $p^{(1)}(x) \in \Delta_L$ denote all probability outputs from model 1 for an $L$-class classification problem. Let $v(p^{(1)}(x)) \in \mathbb{R}^D$ be a list of features extracted from the probability outputs. In all experiments, the post-hoc model is $g(x) = \mathrm{MLP}(v(p^{(1)}(x)))$ with $v$ producing $D = L + 11$ features. These features are

1. The entropy of $p^{(1)}(x)$.

2. Top 10 highest probability values of $p^{(1)}(x)$.

3. One-hot encoding of $\arg\max_{y'} p_{y'}^{(1)}$ (i.e., an $L$-dimensional binary vector).

For the MLP, let $\mathrm{FC}_K$ denote a fully connected layer with $K$ output units without an activation. Let $\mathrm{FC}_{K,f}$ denote a fully connected layer with $K$ output units with $f$ as the activation. In all experiments involving post-hoc rules, we use

$$g(x) = (\mathrm{FC}_1 \circ \mathrm{FC}_{2^4,\mathrm{ReLU}} \circ \mathrm{FC}_{2^6,\mathrm{ReLU}} \circ v \circ p^{(1)})(x),$$

where ReLU denotes the Rectified Linear Unit. Note that both the MLP and the set of input features to $g$ are small. We found that a post-hoc model can easily overfit to its training set. Controlling its capacity helps mitigate this issue.

To train $g$, we use Adam [39] as the optimisation algorithm with a constant learning rate of 0.0007 and batch size 128. For CIFAR 100 experiments, we use a held-out set of size 5000 as the training set. For ImageNet experiments, we use an held-out set of size 10000. We train for 20 epochs. Further, an L2 regularization of weights in each layer in the MLP is also added to the training objective. We set the regularization weight to 0.001.

# D  Calibration Analysis

To further study the performance of confidence-based deferral, Figure 4 presents *calibration plots* [54, 25]. These ideally visualise $\mathbb{P}(y \neq h^{(1)}(x) \wedge y = h^{(2)}(x) \mid x \in \mathcal{X}_q)$ for $q \in [0, 1]$, where $\mathcal{X}_q \doteq \{x \in \mathcal{X} : \max_{y'} p_{y'}^{(1)}(x) = q\}$ is the set of samples where model 1's confidence equals $q$. In practice, we discretise $q$ into 10 buckets.

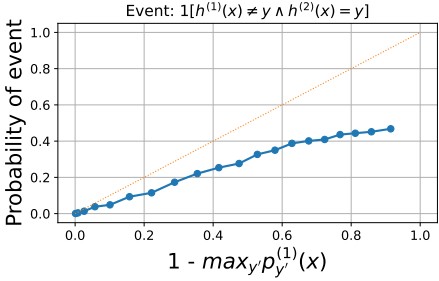
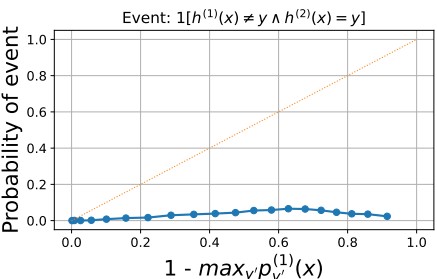

(a) MobileNet V2 and EfficientNet B0 on ImageNet.

(b) MobileNet V2 and **dog specialist** EfficientNet B0 on ImageNet.

Figure 4: Calibration plots visualising the empirical probability of the event $h_1(x) \neq y \wedge h_2(x) = y$, i.e., the first model predicts incorrectly and the second model predicts correctly. In settings where confidence-based deferral performs poorly, the first model's confidence is a poor predictor of whether the second model makes a correct prediction: the likelihood of this event tends to be systematically *over-estimated*.

These plots reveal that, as expected, the confidence of $h^{(1)}$ may be a poor predictor of when $h^{(1)}$ is wrong but $h^{(2)}$ is right: indeed, it systematically *over-estimates* this probability, and thus may result in erroneously forwarding samples to the second model. This suggests using post-hoc estimates of when $h^{(1)}$ is wrong, but $h^{(2)}$ is right.

We remark here that in a different context, Narayan et al. [50] showed that certain cascades of pre-trained language models may be amenable to confidence-based deferral: the confidence of the small model can predict samples where the small model is wrong, but the large model is correct. In this work, we focus on image classification settings. Generalising our analysis to natural langauge processing models will be an interesting topic for future study.

# E  Additional Experimental Results

In this section, we present more experimental results to support our analysis in §4.

## E.1  Confidence-Based Deferral Underperforms Under Label Noise

As illustrated in §5.2, confidence-based deferral can be sub-optimal when there is label noise. The label noise experiment in §5.2 is based on synthetic label noise injected to clean data to have a controlled experiment. The aim of this section is to confirm that we have the same conclusion with a real problem that has label noise. Here, we consider the Mini-ImageNet dataset (with noise rate 60%) from [35]. The two base models in the cascade are two ResNet 10 models with widths 16 (small model) and 64 (large model). Figure 5 shows the results. As in the main text, Diff-01 significantly outperforms Confidence based deferral in this setting.

## E.2  When Model 2 is Highly Accurate

To illustrate that confidence-based deferral is optimal when the second model has a constant error probability, we train an MLP model as $h^{(1)}$ and a CIFAR ResNet 56 as $h^{(2)}$ on the MNIST dataset [43], a well known 10-class classification problem. The MLP has one hidden layer composed of two hidden units with a ReLU activation, and a fully connected layer with no activation function to produce 10 logit scores (for 10 classes). Figure 6 presents our results.

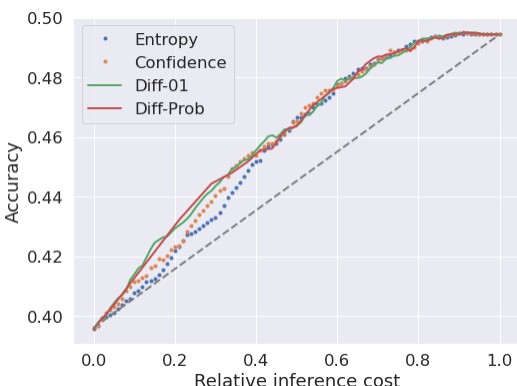

Figure 5: Test accuracy vs deferral rate of the post-hoc approaches (`Diff-01`, `Diff-Prob`), confidence thresholding (Confidence), and entropy thresholding (Entropy). We considered the Mini-ImageNet dataset (with noise rate 60%) from [35]. The two base models are ResNet 10 models with widths 16 (small model) and 64 (large model). Consistent with our analysis, confidence-based deferral underperforms in the presence of label noise.

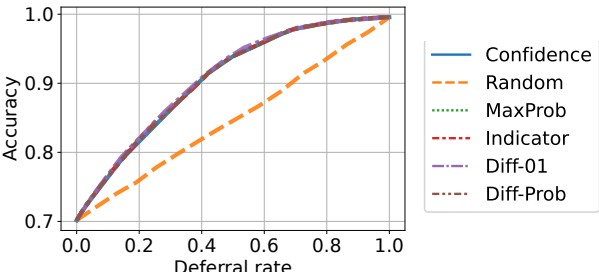

Figure 6: Test accuracy vs deferral rate of post-hoc deferral rules (in Table 1) when the second model is highly accurate. We observe that confidence-based deferral is competitive in this setting.

## E.3 Oracle Curves

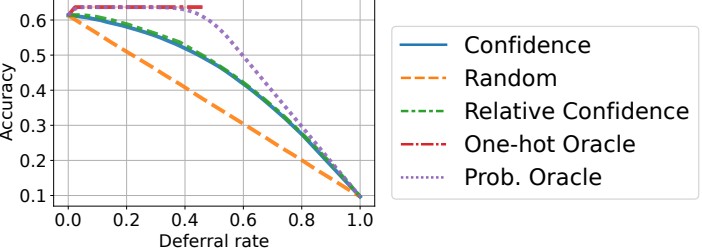

Figure 7: Test accuracy vs deferral rate of different plug-in estimates for the oracle deferral rule in (3) (see §3.3). The first model is trained on all ImageNet classes and the second model specialises on dog classes (i.e., and hence has non-uniform errors). In this case, accounting for the label probability of $h^{(2)}$ (as done by the oracle approaches) can theoretically improve upon confidence-based deferral.

In this section, we illustrate different plug-in estimates for the oracle deferral rule in (3) (see §3.3). We consider two base models: MobileNet V2 [65] and EfficientNet B0 [71] as $h^{(1)}$ and $h^{(2)}$, which are trained independently on ImageNet. The second model is trained on a subset of ImageNet dataset containing only "dog" synset (i.e., a dog specialist model). Figure 7 illustrates the performance of confidence-based deferral along with different plug-in estimates for the oracle deferral rule.

We see that in this specialist setting, confidence-based deferral is sub-optimal, with a considerable gap to the oracle curves; this is expected, since here $\mathbb{P}(y = h^{(2)}(x))$ is highly variable for different $x$.

In particular, $\mathbb{P}(y = h^{(2)}(x)) \sim 1$ for dog images, but $\mathbb{P}(y = h^{(2)}(x)) \sim \frac{1}{L - \#\text{non-dog classes}}$ for other images. There are 119 classes in the dog synset. Note that all Oracle curves are theoretical constructs that rely on the true label $y$. They serve as an upper bound on the performance what we could achieve when we learn post-hoc deferral rules to imitate them.

### E.4 Confidence of a Dog-Specialist Model

In this section, we report confidence of the two base models we use in the ImageNet dog-specialist setting in Figure 7. Recall that $p^{(1)}$ is MobileNet V2 [65] trained on the full ImageNet dataset (1000 classes), and $p^{(2)}$ is EfficientNet B0 [71] trained only on the dog synset from ImageNet (119 classes). We show empirical distributions of $\max_{y'} p_{y'}^{(1)}(x), \max_{y'} p_{y'}^{(2)}(x), \max_{y''} p_{y''}^{(2)}(x) - \max_{y'} p_{y'}^{(1)}(x), p_y^{(2)}(x) - p_y^{(1)}(x)$, and $p_y^{(2)}(x) - p_y^{(1)}(x)$ in Figure 8, grouped by the category of each input image (Dog or Non-Dog). These statistics are computed on the test set of ImageNet dataset, containing 50000 images.

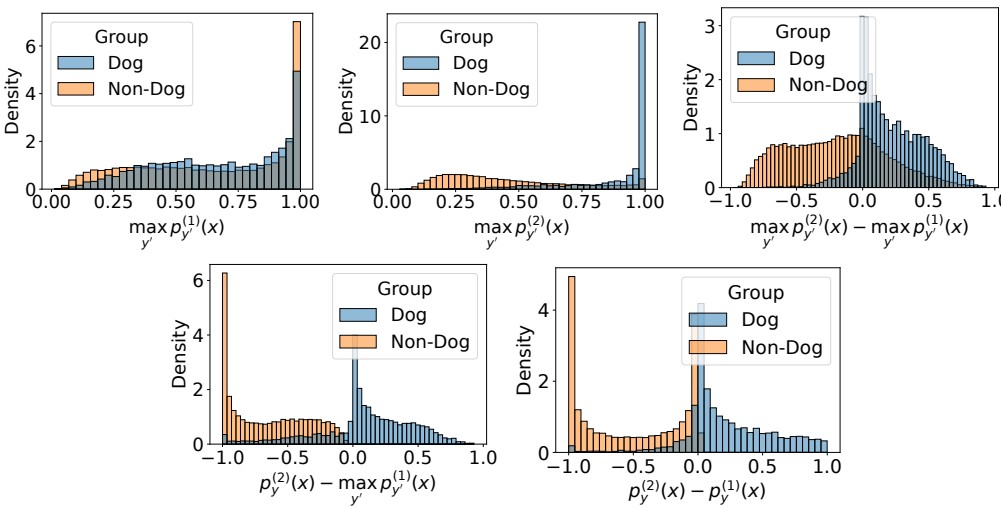

Figure 8: Confidence on ImageNet test set of $p^{(1)}$ (MobileNet V2 trained on the full ImageNet training set), and $p^{(2)}$ (EfficientNet B0 trained only on the dog synset).

We observe from the distribution of $\max_{y'} p_{y'}^{(2)}(x)$ that the specialist EfficientNet B0 is confident on dog images. To the specialist, non-dog images are out of distribution and so $\max_{y'} p_{y'}^{(2)}(x)$ is not a reliable estimate of the confidence. That is, $p^{(2)}$ can be highly confident on non-dog images. As a result, a deferral rule based on Relative Confidence ($\max_{y''} p_{y''}^{(2)}(x) - \max_{y'} p_{y'}^{(1)}(x)$) may erroneously route non-dog images to the second model. Thus, for training a post-hoc model, it is important to consider model 2's probability of the true label (i.e., $p_y^{(2)}(x)$), which is low when $(x, y)$ is a labeled image outside the dog synset.

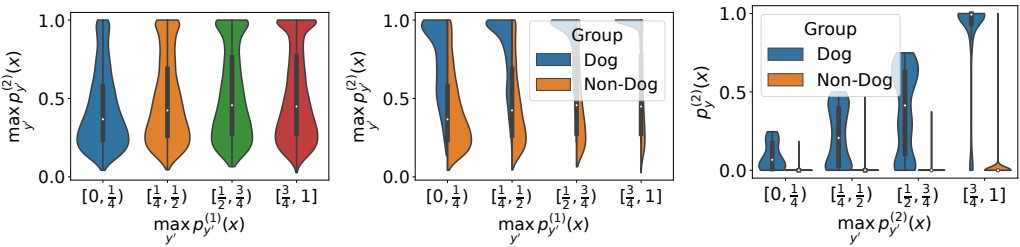

Figure 9: Confidence of MobileNet V2 (trained on the full ImageNet training set) versus confidence of EfficientNet B0 (trained only on the dog synset).

We further show a scatter plot of confidence of both models in Figure 9. We observe that there is no clear relationship between $\max_{y'} p_{y'}^{(1)}(x)$ and $\max_{y'} p_{y'}^{(2)}(x)$.

## F    Extension to Multi-Model Cascades

One may readily employ post-hoc deferral rules to train multi-model cascades. Given $K$ classifiers $h^{(1)}, \ldots, h^{(K)}$, a simple recipe is to train $K - 1$ deferral rules, with the $k$th rule trained to predict whether or not $h^{(k)}$ should defer to $h^{(k+1)}$. Each of these individual rules may be trained with any of the objectives detailed in Table 1. At inference time, one may invoke these rules sequentially to determine a suitable termination point.

In Figure 10, we present results for a 3-model cascade in the label noise setting. As with the 2-model case, post-hoc deferral improves significantly over confidence-based deferral, owing to the latter wastefully deferring on noisy samples that no model can correctly predict. Here, the relative inference cost is computed as the relative cost compared to always querying the large model.

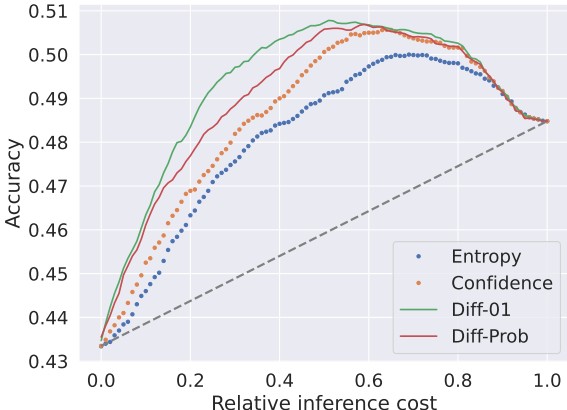

Figure 10: Test accuracy versus deferral rate for a 3-model cascade of ResNet-8, ResNet-14, ResNet-32 . Results are on CIFAR-100 with targetted noise on the first 25 classes. As with the 2-model cascade, post-hoc deferral improves significantly over confidence-based deferral. Here, the relative inference cost is computed as the relative cost compared to always querying the large model.

The above can be seen as a particular implementation of the following generalisation of Proposition 3.1. Suppose we have $K$ classifiers $h^{(1)}, \ldots, h^{(K)}$, with inference costs $c^{(1)}, \ldots, c^{(K)} \in [0, 1]$. We assume without loss of generality that $0 = c^{(1)} \leq c^{(2)} \leq \ldots \leq c^{(K)}$, i.e., the costs reflect the *excess* cost over querying the first model.

Now consider learning a *selector* $s \colon \mathcal{X} \to [K]$ that determines which of the $K$ classifiers is used to make a prediction for $x \in \mathcal{X}$. Our goal in picking such a selector is to minimise the standard misclassification accuracy under the chosen classifier, plus an appropriate inference cost penalty:

$$R(s; h^{(1)}, \ldots, h^{(K)}) \doteq \mathbb{P}(y \neq h^{(s(x))}(x)) + \mathbb{E}[c^{(s(x))}]. \tag{11}$$

We have the following, which is easily seen to generalise Proposition 3.1.

**Lemma F.1.** *The optimal selector for* (11) *is given by*

$$s^*(x) = \underset{k \in [K]}{\operatorname{argmin}} \, \mathbb{P}(y \neq h^{(k)}(x)) + c^{(k)}. \tag{12}$$

*Proof of Lemma F.1.* Observe that

$$R(s; h^{(1)}, \ldots, h^{(K)}) = \mathbb{P}(y \neq h^{(s(x))}(x)) + \lambda \cdot \mathbb{E}c^{(s(x))}$$
$$= \mathbb{E}_x \mathbb{E}_{y|x} 1(y \neq h^{(s(x))}(x)) + \lambda \cdot c^{(s(x))}.$$

Now suppose we minimise $s$ without any capacity restriction. We may perform this minimisation pointwise: for any fixed $x \in \mathcal{X}$, the optimal prediction is thus

$$s^*(x) = \operatorname*{argmin}_{k \in [K]} \mathbb{E}_{y|x} 1(y \neq h^{(k)}(x)) + \lambda \cdot c^{(k)}.$$

$\square$

When $K = 2$, we exactly arrive at Proposition 3.1. For $K > 2$, the optimal selection is the classifier with the best balance between misclassification error and inference cost.

Lemma F.1 may be seen as a restatement of Trapeznikov and Saligrama [74, Theorem 1], which established the Bayes-optimal classifiers for a sequential classification setting, where each classifier is allowed to invoke a "reject" option to defer predictions to the next element in the sequence. This equivalently packs together a standard classifier $h^{(k)}$ and deferral rule $r^{(k)}$ into a new classifier $\bar{h}^{(k)}$.

## G Limitations

In this work, we have identified problem settings where confidence-based deferral can be sub-optimal. These problem settings are (1) specialist models, (2) label noise, and (3) distribution shift (see Figure 2 for experimental results in these settings). These problem settings are not exhaustive. Identifying other conditions under which confidence-based deferral performs poorly is an interesting direction for future work. Another interesting topic worth investigating is finite-sample behaviors of cascades, both with confidence-based deferral and with post-hoc rules.