# OpenReview forum: "When Does Confidence-Based Cascade Deferral Suffice?"
_NeurIPS.cc/2023/Conference — NeurIPS 2023 poster_

### Official Review · Reviewer_ahRx · 2023-06-23

**Soundness:** 2 fair
**Presentation:** 2 fair
**Contribution:** 2 fair
**Rating:** 7
**Confidence:** 4

**Summary:**

The paper consists of 2 parts. Part 1 contains theoretical analysis of when confidence-based deferral rules for cascades of 2 or more models succeed or fail, based on a proposed risk function (equation (1) in section 3) presenting a tradeoff between accuracy and computational cost of invoking subsequent models. Part 2 proposes new deferral rules that are more sophisticated than existing confidence-based deferral rules using a machine-learned postdoc model. Experiments attempt to justify claims made in part 1 and to show that posthoc-based deferral rules produce better results than others.

**Strengths:**

The paper provides a number of edge cases, both good and bad, together with good explanations on those edge cases.
References to previous works are excellent. However, I cannot claim my expertise in this area.

**Weaknesses:**

I have found a few places that appear rather puzzling in the followings:

1. Lemma 4.1 line 176: it is unclear what authors mean by "produce the same ordering over instances x". The lack of a precise formula makes it unnecessarily difficult to follow the proof for lemma 4.1 and to verify the proof, because I did not know what end result to expect. Eventually I understood what that term meant after carefully checking the appendix. But it was via a few definitions in the proof in the appendix, but not in the lemma statement itself. The statement definitely needs revising.

2. Section 4.1 seems to have a technical flaw in reasoning, despite that Lemma 4.1 appears to be correct. Consider functions $R1(c, c^{(1)}) = R(r_{conf} | c, c^{(1)})$ and $R2(c) = R(r^* | c)$. Lemma 4.1. shows a condition when the accuracy-deferral curve of $r^*$ matches with that of $r_{conf}$. But that is an analysis with respect to accuracy, not risk $R(\cdot)$ defined in (1). For a given $c$, for example $c$ represents a trade off between accuracy and inference speed, $R2(c)$ is constant but Lemma 4.1 does not guarantee that the lowest value of $R1(c, c^{(1)})$ over $c^{(1)}$ cannot go lower than $R2(c)$. To me, it only makes sense if we assume that the algorithm sees $c$ as a variable and we consider $c^*, R^* = \[arg\]min_c{R2(c)}$ instead. However, in such cases there are 2 issues: (1) Lemma 4.1 still does not guarantee that across all possible $(c, c^{(1)})$ pairs, the only time when $R1 = R^*$ is when $c=c^*$, and (2) if $c$ is a variable then how do you take practical considerations like inference speed into the analysis?
While the subsequent success/failure cases in 4.2 seem intuitive to me, I just don't see how Lemma 4.1 can be used to convincingly explain them.

3. The general idea proposed in 4.2 is somewhat weak as well. In order to make a deferral rule for a cascade of 2 models to work better, you need to create a third model, i.e. the posthoc model g(x)? Wouldn't that incur an additional computational cost, meaning the risk function in (1) has to be redefined? I do not see any quantisation of that effect in the experiments.
In addition, wouldn't introducing g(x) also mean that you have given a bit of training power to complement model 1 and 2, whereas the general setting of the paper is to treat model 1 and 2 as pre-trained blackboxes not to be tampered with (e.g. line 244-246)?

4. Section 5 presents all results in terms of accuracy-deferral curves. However, I cannot see any curve or any piece of information at all mentioning the risk $R(\cdot)$ in (1). This is puzzling. On one hand, sections 3 and 4 discuss minimising risk in (1) which can be seen as a tradeoff between accuracy and computational cost. On the other hand, all experiments only present information related to accuracy. I am not sure how those results can completely justify sections 3 and 4.

**Questions:**

I need to be able to see that the paper has an objective and that every section is consistent towards that objective. At the moment, the three sections 3, 4 and 5 do not appear to be completely aligned with each other. If you could convince me where I am wrong in the Weaknesses section above, and show me some form of consistency throughout the paper, I will change my opinion.
It is unclear to me whether the constant cost $c$ on line 117 in (1) is to be treated as constant or not. It appears to be constant at first (and it makes sense to do so from a practical point of view). But then in section 4.1, $c$ seems to be treated as a variable that can be optimised away. What is the point of having an accuracy-deferral curve for $r^*$ over $c$ if $c$ is constant?

**Limitations:**

There is no societal negative impact from the paper.

---

> ### Author Rebuttal · Authors · 2023-08-10
>
> > Section 4.1 seems to have a technical flaw in reasoning, despite that Lemma 4.1 appears to be correct… Section 5 presents all results in terms of accuracy-deferral curves… I cannot see any curve or any piece of information at all mentioning the risk $R(\cdot)$ in (1)
>
> **There appears to be a misunderstanding which we’d like to clarify**. We believe the reviewer’s doubt arises since:
> Section 3 considers the cascade _risk_ (Equation 1) at a fixed _cost parameter_ $c$
> Section 4 and 5 considers the cascade _accuracy_ at a fixed _deferral rate_ $\tau$, as noted in L126 and L177
>
> These respectively yield cost-risk curves (as $c$ is varied) and deferral curves (as $\tau$ is varied). These are _equivalent_ ways of assessing the overall quality of a cascade (see below), and so we use them interchangeably. [This is analogous to the equivalence between ROC and cost curves in classification.] Following prior work (Bolukbasi et al., 2017), (Kag et al., 2023), our experiments use deferral curves to compare methods. We will make this clearer.
>
> In more detail:
>
> Section 4 seeks to identify when confidence-based deferral produces the “best” possible cascade. By “best”, we mean the cascade which achieves the _best possible deferral curve_, i.e., which achieves maximal accuracy at *all* deferral rates. **Equivalently** (see below), this is the cascade which achieves the best possible _cost-risk curve_, i.e., which achieves minimal risk (1) for *all* cost parameters $c$.
>
> To make progress, we characterise the “best” cost-risk curve in Proposition 3.1. This shows that to achieve the “best” curve as $c$ varies in $[0, 1]$, we can threshold $s^*( x ) = \eta\_{h^{(2)}(x)}(x) - \eta\_{h^{(1)}(x)}(x)$ by $c$.
>
> Crucially, there is a **one-to-one correspondence** between the optimal cost-risk and deferral curves. Thus, Proposition 3.1 also characterises the optimal deferral curve, which we reference in Lemma 4.1. To see this correspondence, recall that $\forall c \in [0, 1]$, the risk (1) is
>
> $$ R(r; h^{(1)}, h^{(2)}) = \mathbb{P}(y \neq h^{(1)}(x), r(x)=0) +
>  \mathbb{P}(y \neq h^{(2)}(x), r(x)=1) + c \cdot \mathbb{P}(r(x)=1). $$
>
> This captures
>
> $$ (\text{classification error of the cascade}) + c \cdot (\text{deferral rate}). $$
>
> Appealing to the Lagrangian — analogous to the Neyman-Pearson lemma in ROC analysis — $\exists \tau \ge 0$ such that minimising $R(r; h^{(1)}, h^{(2)})$ over $r$ is equivalent to
>
> $$ \max_{r}  \quad (\text{classification accuracy of the cascade}) \quad \text{subject to} \quad  (\text{deferral rate}) \leq \tau. $$
>
> Thus, tracing out the optimal cost-risk curve for varying $c$ will trace out the optimal deferral curve for varying $\tau$.
>
> In fact, the two curves have an intimate **point-line duality**. Any point $(d_0, e_0)$ in deferral curve space – where $d$ denotes deferral rate, and $e$ denotes error – can be mapped to the line $\{ (c, c \cdot d_0 + e_0) : c \in [0, 1] }$ in cost curve space. Conversely, any point $(c_0, r_0)$ in cost curve space – where $c_0$ denotes cost, and $r_0$ denotes risk – can be mapped to the line $\{ (d, r_0 - c_0 \cdot d) : d \in [0, 1] \}$ in deferral curve space. This is analogous to the correspondence between ROC and cost-risk curves in classification (Drummond & Holte, “Cost curves: An improved method for visualizing classifier performance”, ‘06).
>
> **We are happy to answer any further queries.**
>
> > What is the point of having an accuracy-deferral curve for $r^*$ over $c$ if $c$ is constant?
>
> Optimising (1) may be cast as a constrained optimisation problem where $c$ (or its re-parameterisation) controls the deferral rate. Changing $c$ changes the constrained optimisation problem. It is thus natural that its solution $r^*$ depends on $c$. Intuitively, varying $c$ allows one to trade off accuracy and inference cost. In practice, one will be operating at a particular point on this trade-off curve, and so there is only one fixed $c$ at inference time.
>
> The deferral curve helps compare two deferral rules **over all possible choices of the deferral rate**. This is analogous to the use of ROC curves to compare two classifiers over all possible values of relative false positive to false negative cost.
>
> >  cost of the posthoc model g(x)
>
> __Our post-hoc model $g$ is a simple feedforward network and is much smaller than the two base models__. This is described in **Section C.2 (Appendix)** where $g$ has no more than 100 hidden units. In comparison, on the ImageNet problem, we use MobileNet v2 as $h^{(1)}$ (3.5 million parameters), and EfficientNet B0 as $h^{(2)}$ (5.3 million parameters). The cost of invoking $g$ at inference time is negligible with respect to these models. We are happy to add a comment noting this.
>
>
> > [by introducing g(x)] you have given a bit of training power to complement model 1 and 2, whereas the general setting of the paper is to treat model 1 and 2 as pre-trained blackboxes?
>
> We indeed consider the setting where one cannot modify the models $h^{(1)}$ and $h^{(2)}$. However, exploiting the models’ _outputs_ for further learning is perfectly admissible.
>
> Per L244-246, we are allowed to train a separate post-hoc deferral model $r$. When we use the risk $R(r; h^{(1)}, h^{(2)})$ (see (1)), we optimise for $r$ which is parameterised as thresholding $g(x)$.
>
>
> >  Sections 3, 4 and 5 do not appear to be completely aligned
>
> Our work studies when confidence-based deferral suffices. The logical flow is:
> In Section 3, we characterise the form of the optimal deferral rule.
> In Section 4, we identify specific conditions under which confidence-based deferral may disagree with the optimal rule, and study post-hoc deferral approaches in an attempt to address these limitations.
> In Section 5, we present experiments that verify the above failure modes.
>
> > Meaning of "produce the same ordering over instances x"
>
> We apologise for not being clearer here. Please see the global response.

---

> > ### Comment · Reviewer_ahRx · 2023-08-14
> > **Thank you. Points 1, 2 and 4 cleared. Point 3 remains for discussion.**
> >
> > Thank you for the clarifications.
> >
> > >> Section 4.1 seems to have a technical flaw in reasoning, despite that Lemma 4.1 appears to be correct…
> >
> > > There appears to be a misunderstanding which we’d like to clarify...
> >
> > Right. The Lagrangian manipulation to connect between the unconstrained risk in (1) and accuracy with a constraint. Clearly, without this fact it is difficult to link section 3 with sections 4 and 5. This is a crucial missing bit, which clears both my points 2 and 4. Why did it not appear in the original version? It would have been a lot easier to read had the risk-accuracy equivalence got established in section 3.
> >
> > Since points 2 and 4 are major, I am ready to increase my ratings by the way.
> >
> > >>  cost of the posthoc model g(x)
> >
> > > Our post-hoc model is a simple feedforward network and is much smaller than the two base models...
> >
> > I understand your argument. But your paper's title is `When Does Confidence-Based Cascade Deferral Suffice?`. To a normal reader, the title suggests that the cascades of interest contain pretrained models plus some thresholding on the model confidences, no additional training is needed. A posthoc model would require the user to train additionally. Under a research context this may be fine. But in real world, this additional requirement can be costly due to resources and teams involved. It may be better to improve the pretrained models instead.
> >
> > I am not saying the idea of using a post-hoc model is wrong. I am saying that in my practical view point, it somewhat backfires.
> >
> > In addition, a hidden assumption here is that the posthoc model g(x) must be very lightweight compared to the pretrained models. But in very high speed applications where the pretrained models can be marginally slower than the posthoc model g(x) then this assumption will break.

---

> > > ### Author Response · Authors · 2023-08-14
> > > **We seek to understand when confidence-based deferral may fail, and when alternate deferral strategies can perform better**
> > >
> > > We thank the reviewer for their detailed feedback.
> > >
> > > > Right. The Lagrangian manipulation to connect between the unconstrained risk in (1) and accuracy with a constraint. Clearly, without this fact it is difficult to link section 3 with sections 4 and 5. This is a crucial missing bit, which clears both my points 2 and 4. Why did it not appear in the original version? It would have been a lot easier to read had the risk-accuracy equivalence got established in section 3.
> > >
> > > We are glad this clarifies! We apologise if this point wasn’t clear in the original submission. In our revised version, **we will expand the para in L126 - L130** to explicate the Lagrangian view of (1), and the equivalence of the cost-risk and deferral rate-accuracy curves.
> > >
> > >
> > > > I understand your argument. But your paper's title is When Does Confidence-Based Cascade Deferral Suffice?. To a normal reader, the title suggests that the cascades of interest contain pretrained models plus some thresholding on the model confidences, no additional training is needed.
> > >
> > > Our intended logical flow is the following. First, our analysis in Section 3 and 4.1 precisely studies conditions under which confidence-based cascades may _not_ suffice (e.g., label noise). This is in line with the title.
> > >
> > > Next, given that there are conditions where such cascades _don’t_ suffice, a natural question is whether some alternate strategy _does_. To that end, Section 4.2 proposes and studies post-hoc models, which are a minimal and (in our opinion) natural extension.
> > >
> > > We attempted to convey this framing in the writing (e.g., L8 “In this paper, we seek to better understand the conditions under which confidence-based deferral may fail, and when alternate deferral strategies can perform better”). Nonetheless, we are certainly happy to update the text if the reviewer believes this is not made sufficiently clear.
> > >
> > > > A posthoc model would require the user to train additionally. Under a research context this may be fine. But in real world, this additional requirement can be costly due to resources and teams involved… In addition, a hidden assumption here is that the posthoc model g(x) must be very lightweight compared to the pretrained models. But in very high speed applications where the pretrained models can be marginally slower than the posthoc model g(x) then this assumption will break.
> > >
> > > We certainly agree that there can be settings where post-hoc models may not be appropriate. Nonetheless, we argue that **they are many practical settings where they _are_ appropriate**. As a remark, per L246, we note that prior work such as [31, 44, 68] also considered the use of auxiliary models to improve cascading.
> > >
> > > In our view, cascading makes sense when at least one constituent model is highly compute-intensive (both for inference _and_ training). Given this, _if_ the post-hoc model is significantly lightweight compared to the smallest constituent model, it adds minimal extra overhead both for training and inference.
> > >
> > > Now, if one or more of the constituent models is itself lightweight, then the reviewer is correct that the post-hoc model may add an overhead compared to regular confidence-based cascading. However, if there is a sufficiently large cost gap between the smallest and largest model in the cascade, then the post-hoc approach _may still offer a more favourable cost-quality tradeoff_ than regular confidence-based cascading (which, per Section 4.1, may underperform in some settings).
> > >
> > > The reviewer is completely correct that in some settings, alternate strategies (e.g., improving the base models) may be appropriate. Our purview however is to understand the space of strategies for combining multiple models via a deferral mechanism.

---

> > > > ### Comment · Reviewer_ahRx · 2023-08-15
> > > >
> > > > > We attempted to convey this framing in the writing (e.g., L8 “In this paper, we seek to better understand the conditions under which confidence-based deferral may fail, and when alternate deferral strategies can perform better”). Nonetheless, we are certainly happy to update the text if the reviewer believes this is not made sufficiently clear.
> > > >
> > > > I am glad we have aligned on this point. Yes, I believe it was not very clear earlier.
> > > >
> > > > > We certainly agree that there can be settings where post-hoc models may not be appropriate. Nonetheless, we argue that they are many practical settings where they are appropriate...
> > > >
> > > > Again, I am seeking for clarity in the text here. Yes, I understand there are settings where the post-hoc models are appropriate and there are settings where the post-hoc models are not appropriate. It was just not very clear in the original version which settings were the post-hoc models appropriate. For example, you did not clearly mention that the post-hoc models needed to be significantly faster than the second pretrained model. Otherwise, the cost $c$ that the user had to choose could have been affected by the introduction of a posthoc model, if $c$ had taken inference speed into account. A simple sentence in the main text could have resolved this.
> > > >
> > > > That said, this is a minor point. I have updated my score to 7: Accept.

---

> > > > > ### Author Response · Authors · 2023-08-15
> > > > >
> > > > > Thanks again for the comments.
> > > > >
> > > > > > For example, you did not clearly mention that the post-hoc models needed to be significantly faster than the second pretrained model... A simple sentence in the main text could have resolved this.
> > > > >
> > > > > Thanks for the suggestion -- we agree that stating this point explicitly would make things clearer! We will update the text in Section 4.2 (specifically, the para starting L211) based on the discussion.

---

### Official Review · Reviewer_V2z7 · 2023-07-05

**Soundness:** 3 good
**Presentation:** 4 excellent
**Contribution:** 3 good
**Rating:** 6
**Confidence:** 3

**Summary:**

This paper explores the cascade deferral problem, which is an issue in the context of machine learning models arranged in a cascading order, where a decision needs to be made about whether to defer the processing of data from one model to the next in the cascade. The challenge is to optimize the deferral decision to achieve the best trade-off between computational costs and model performance.

Previously, this problem was often addressed using confidence-based deferral rules. This approach involves deferring data to the next model in the cascade when the first model's confidence in its prediction falls below a certain threshold. However, the paper identifies limitations with the confidence-based deferral method. Specifically, it may underperform in certain settings, such as when the second model in the cascade is a specialist model, or when there's label noise in the training data.

To overcome these limitations, the authors introduce post-hoc deferral rules. Unlike the confidence-based approach, post-hoc deferral rules use additional information, beyond just the confidence of the first model, to make deferral decisions. These rules are trained and optimized to provide better accuracy-cost trade-offs.

The authors compare the performance of confidence-based and post-hoc deferral rules under various experimental settings. They use datasets like ImageNet and CIFAR 100, with settings including a specialist model scenario, label noise, and distribution shift. They find that post-hoc deferral significantly outperforms confidence-based deferral in scenarios involving a specialist second model and label noise. However, they also identify potential overfitting issues with post-hoc deferral, highlighting the need for careful capacity control.


**Strengths:**

**Originality**: The paper introduced a post-hoc deferral scheme that utilizes a Bayes-optimal deferral rule. In particular, it addresses issues not resolved by the traditional confidence-based deferral methods.

**Quality**: The authors effectively utilize mathematical proofs and models to construct and explain their deferral schemes, and they validate these models through extensive experiments on established datasets such as ImageNet and CIFAR 100.

**Clarity**: The paper is well-structured and clearly written. The paper's theoretical concepts are explained with clarity and are substantiated with illustrative figures, while the experimental design and results are presented in detail.

**Significance**: The research addressed a critical limitation in the commonly used confidence-based deferral schemes. Confidence-based methods, while widely used, tend to underperform in certain situations such as in specialist settings or when there is label noise. This research introduces and investigates the efficacy of post-hoc deferral models as a solution to this problem, offering a more optimal and efficient method of deferral. The paper's analysis of the limitations of the confidence-based methods and the demonstration of how the post-hoc deferral models overcome these issues are essential for further research in this area.

**Weaknesses:**

**Limited Dataset and Task Diversity**: The empirical results of this paper are primarily based on two datasets: CIFAR-100 and ImageNet. Additionally, the experiments were conducted on image classification tasks only. Expanding the scope of datasets and including other tasks, such as object detection, segmentation, or even venturing into different domains like natural language processing or audio processing, could have provided a more comprehensive evaluation of the post-hoc deferral models.

**Generalizability**: The authors point out that post-hoc models can overfit and fail to generalize well, even when controlling the capacity of the model.


**Questions:**

The authors acknowledge that post-hoc models can overfit and struggle to generalize. Could the authors provide some intuitive explanation for the reasoning behind this issue?

**Limitations:**

The authors have adequately addressed the limitations

---

> ### Author Rebuttal · Authors · 2023-08-10
>
> We thank the reviewer for carefully reviewing our work, and for the complete summary which we agree with.
>
> >  The empirical results of this paper are primarily based on two datasets: CIFAR-100 and ImageNet. Additionally, the experiments were conducted on image classification tasks only.
>
> In the present work, we aim to illustrate when confidence-based deferral can fail with classification  problems. The Bayes optimal deferral rule we derive in Proposition 3.1 and subsequent analyses all start from the risk in Eq (1), which is based on the 0-1 loss. Specifically, the risk of a deferral rule $r$ can be written as
>
> $$ R(r; h^{(1)}, h^{(2)}) = \text{const.} + \mathbb{E}\_{x} 1[r(x)=1] \mathbb{E}\_{y|x} \left[1[y \neq h^{(2)}(x)]-1[y\neq h^{(1)}(x)]+c\right]. $$
>
> Note that the risk depends on 0-1 losses of the two base models $h^{(1)}, h^{(2)}$. **For other tasks, one can simply replace these base losses to other appropriate losses (e.g., log loss for language models) and the Bayes optimal rule can be derived.** We plan to include results on NLP classification tasks in the revised version. Please note that we have added results on Mini-ImageNet to the PDF as part of this rebuttal. Please see the response to Reviewer kqwH for details.
>
>
> >  The authors point out that post-hoc models can overfit and fail to generalize well, even when controlling the capacity of the model. Could the authors provide some intuitive explanation for the reasoning behind this issue?
>
> The studied post-hoc approaches are not without limitations. It can be hard for a small post-hoc model to reliably predict the confidence or confidence difference of much larger models (underfitting). The overfitting case arises especially in (but not limited to) the generalist setting. In this setting, the two base models $h^{(1)}, h^{(2)}$ perform roughly equally well, and there is no obvious set of instances that should be deferred to either of them. In this case, the learned post-hoc deferral model may fail to generalise. This is in contrast to, say, the specialist setting where there is a clear set of instances that should be deferred to $h^{(2)}$ (i.e., the data sub-group that model 2 specialises on).

---

### Official Review · Reviewer_6Hfw · 2023-07-07

**Soundness:** 4 excellent
**Presentation:** 3 good
**Contribution:** 3 good
**Rating:** 7
**Confidence:** 3

**Summary:**

This paper systematically investigates why and when the confidence-based deferral rule work, and particularly, identifies cases when it fails. To enable this investigation, they provide a theoretical characterization of the problem and the optimal deferral rule. They also provide a post-hoc solution that can work well in cases when the confidence-based method fails.

**Strengths:**

The paper is clearly written.
The paper provides a nice theoretical framework for studying the deferral rule in general.
The paper provides lots of useful insights, supported by both theoretical and empirical evidence.


**Weaknesses:**

The listed conditions about when the confidence-based method fails are not that surprising, this leads one wondering the significance of such a contribution if the findings are already something intuitive that need not proof. Anyhow, this might not count as a weakness, as it is also reassuring that the theory does produce intuitively sensible results.

**Questions:**

This deferral rule is about selecting models, while there is also this line of research about selecting samples for the noisy data case. Basically, for the data with noisy labels, one would want to select only the clean ones to do the training, and there are also many confidence-based methods to do this selection. I am wondering how these two lines of research connect with each other, and can you similarly identify cases when the confidence-based method fails in label denoising?

It would be good if the authors can provide some discussion/ideas about how to identify whether it is good to use the confidence-based method in practice. Basically, is there a unified way to identify the failed cases from the data, instead of checking if the data have the issues related to each failed case?

**Limitations:**

The authors have addressed the limitations.

---

> ### Author Rebuttal · Authors · 2023-08-10
>
> > The listed conditions about when the confidence-based method fails are not that surprising. Anyhow, this might not count as a weakness, as it is also reassuring that the theory does produce intuitively sensible results.
>
> We are glad the reviewer finds the results intuitive. We would like to emphasise that despite cascades being studied in various forms for several decades, there has been limited study of the conditions under which confidence-based cascading suffices as a deferral mechanism. We would hope that by clearly laying out specific conditions under which this technique does not suffice, we can aid practitioners in their decision of whether or not to invest in more complex modelling strategies (e.g., post-hoc approaches). This is particularly true for the case of label noise, which we expect to be encountered in many (though not all) practical settings.
>
>
> > .. for the data with noisy labels, one would want to select only the clean ones to do the training, and there are also many confidence-based methods to do this selection. I am wondering how these two lines of research connect with each other
>
> We thank the reviewer for mentioning this broadly related topic of noisy label learning. Please note that label denoising is outside the scope of the present work. In our work, we address the setting where the two (or more generally $K > 2$) base models are already trained and fixed. The goals are to investigate the performance of confidence-based deferral, identify conditions under which it can fail, and identify approaches to learn a post-hoc rule to remedy it.  In our experiments with label noise (i.e., second row of Figure 2), we assume that the given base models were trained with label noise without an option of fine-tuning them on clean labels.
>
> We are happy to provide a connection if the reviewer could please point out which work the reviewer has in mind for confidence-based methods for label denoising.
>
>
> >  how to identify whether it is good to use the confidence-based method in practice… is there a unified way to identify the failed cases from the data,
>
> This is an interesting and important question. We think that the simplest approach is to use a held-out validation dataset that has the same distribution as the test data, produce deferral curves of both confidence-based deferral and post-hoc deferral approaches, and compare them.
> The three failure cases identified in our work are for studying sufficient causes that lead to a failure of confidence-based deferral. In practice, for the purpose of deciding whether to deploy confidence-based deferral, there is no need to exactly identify one of the three settings.

---

> > ### Comment · Reviewer_6Hfw · 2023-08-16
> >
> > I have read the rebuttal, I am quite satisfied and have no further questions.

---

### Official Review · Reviewer_kqwH · 2023-07-19

**Soundness:** 3 good
**Presentation:** 3 good
**Contribution:** 3 good
**Rating:** 7
**Confidence:** 4

**Summary:**

The authors present a theoretical analysis for the Bayes optimal deferral rule for a cascade of K=2 classifiers and a certain population risk.
Based on this rule, they characterize when confidence deferral rule using exact posterior probabilities is similar to the Bayes optimal deferral rule is some sense.
Based on their analysis, the authors (informally) discuss cases where (practical) confidence deferral rule may or may not be sufficient, and proposed several post-hoc methods for the latter.
The insights and the methods are examined empirically.




**Strengths:**

I find the paper very interesting and the analysis to be solid.
The theory is important and the insights and practical guidance that it provides look useful.

**Weaknesses:**

The empirical coverage may be somewhat improved, e.g., by using practical datasets without the controlled modifications (such as the dog-specific classifier). I also wonder whether there is some benchmark that can be used for comparing the new post-hoc methods to existing ones.

**Questions:**

Please improve the explanation for Lemma 4.1.
State what you mean in "the same ordering" in the main body of the paper and not only in the proof in the appendix.

The text below Lemma 4.1, and actually in several other places in the paper, needs to be more accurate when you claim optimality of confidence-based deferral rule based on your analysis --- as by saying so, you assume that the classifier's softmax values are exactly the true posterior probabilities, which is obviously not the case in practice [21].

In Figure 2, first row, how do you explain the fact that plain confidence deferral oftentimes outperforms the two post-hoc methods Diff-01 and Diff-prob? Furthermore, Diff-prob in fact is consistently worse than the other methods in all 3 rows in Figure 2, can you explain why?

A minor comment: In Algorithm 1, when you write predict inside the loop, letting it be followed by a "break" command would make the scheme clearer.

**Limitations:**

---

> ### Author Rebuttal · Authors · 2023-08-10
>
> Thanks for the positive feedback, and for recognizing the importance of our theoretical analysis.  We appreciate the reviewer’s comments on the presentation. We will revise accordingly.
>
>
> > The empirical coverage may be somewhat improved, e.g., by using practical datasets without the controlled modifications (such as the dog-specific classifier).
> Thanks for the excellent suggestion. **We have confirmed that we obtain similar conclusions on a dataset with realistic label noise.**
>
>
> We considered the Mini-ImageNet dataset (with noise rate 60%) from Jiang et al., Beyond Synthetic Noise: Deep Learning on Controlled Noisy Labels, ICML ‘20. In the rebuttal PDF, we have included deferral curves for various methods under a cascade of ResNet 10 models with widths 16 (small model) and 64 (large model). As in the body, Diff-01 significantly outperforms Confidence based deferral in this setting.
>
> >   Text below Lemma 4.1: you assume that the classifier's softmax values are exactly the true posterior probabilities,
>
> To clarify, Lemma 4.1 and text below it **does not** assume that the first model’s softmax probabilities are exactly the true posterior probabilities. Rather, the key assumption is that we accurately estimate the **error probability** of model 1, which is a weaker condition.
>
> In more detail:
>
> Recall that $h^{(i)}$ denotes the i-th classifier where $i \in \\{1, 2\\}$. In line 153, we define $\eta_{h^{(i)}(x)}(x) := \mathbb{P}(y=h^{(i)}(x) | x)$. This can be interpreted as the _conditional_ accuracy of the classifier $h^{(i)}$ on an instance $x$. Intuitively, this quantity captures the agreement between the prediction $h^{(i)}$ and that of the Bayes’ classifier $\mathbb{P}(y | x)$ on $x$. The Bayes optimal deferral rule $r^{*}(x) =\boldsymbol{1}\left[\eta_{h^{(2)}(x)}(x)-\eta_{h^{(1)}(x)}(x)>c\right]$ given in Proposition 3.1 means that we should defer $x$ when the relative gain in conditional accuracy is larger than $c$.
>
> Lemma 4.1 characterises when confidence-based deferral (i.e., based on only $\eta_{h^{(1)}(x)}(x)$) is optimal. While measuring the confidence with $\eta_{h^{(1)}(x)}(x)$ is ideal, in practice, we do not have access to it since it depends on the true conditional distribution $\mathbb{P}(y | x)$. The commonly used confidence measure  $\max\_{y'} p^{(1)}\_{y'}( x )$ may be regarded as an estimator of  $\eta\_{h^{(1)}(x)}(x)$.
>
> A practical interpretation of Proposition 3.1 and Lemma 4.1 would require us to assume that $\max\_{y'} p^{(1)}\_{y'}( x )$ is close to $\eta_{h^{(1)}(x)}(x) = \mathbb{P}( y \neq h^{(1)}( x ) \mid x )$. In other words, the confidence $\max\_{y'} p^{(1)}\_{y'}( x )$ captures the conditional accuracy. However, this is not the same as stating that the classifier's softmax values are exactly the true posterior probabilities. The latter translates to $p_y^{(1)}(x) = \mathbb{P}(y | x)$ for $x \in \mathcal{X}, y \in [L]$, a strong condition that we do not assume. In fact, assuming so would have defeated the purpose of forming a cascade, since model 1 alone suffices.
>
> > In Figure 2, first row, how do you explain the fact that plain confidence deferral oftentimes outperforms the two post-hoc methods Diff-01 and Diff-prob
>
> As described in Section 4.1 (lines 187-195), when $h^{(2)}$ is a specialist, it performs exceptionally well on a data sub-group $\mathcal{X}_{\mathrm{good}}$ that it specialises on, and performs poorly on the rest of the data $\mathcal{X}\_{\mathrm{bad}} = \mathcal{X}  \setminus \mathcal{X}\_{\mathrm{good}}$. In other words, $\eta\_{h^{(2)}(x)}(x)$ is high on instances in $\mathcal{X}\_{\mathrm{good}}$ and low on instances in $\mathcal{X}\_{\mathrm{bad}}$.
>
> A post-hoc deferral rule can perform well by learning to defer $x \in \mathcal{X}\_{\mathrm{good}}$ to $h^{(2)}$ and defer $x \in \mathcal{X}\_{\mathrm{bad}}$  to $h^{(1)}$. That is, the post-hoc rule can learn to identify the set of instances $x$’s for which $\eta\_{h^{(2)}(x)}(x) - \eta\_{h^{(1)}(x)}(x) > c$ (recall the optimal deferral rule in Proposition 3.1).  This is the case in Figure 2(c) where $h^{(2)}$ is a dog specialist model.
>
> However, in Figure 2(b), the fraction of non-dog training images for $h^{(2)}$ is 4% (compared to 2% in Figure 2(c)). In this case, $h^{(2)}$ becomes relatively more like a generalist, and  $\eta\_{h^{(2)}(x)}(x) - \eta\_{h^{(1)}(x)}(x)$ is closer to 0 i.e., less accuracy gap between the two base models. This makes it harder for a post-hoc deferral rule to learn to identify instances to defer. This is why Confidence can be better than post-hoc rules in this case.
>
>
> > Diff-prob in fact is consistently worse than the other methods in all 3 rows in Figure 2, can you explain why
>
> This is an interesting question. While we don’t have a conclusive answer, our current hypothesis is that it is the strong assumption of Diff-Prob that causes such poor performance.
>
> To start, recall the Bayes optimal deferral rule from Proposition 3.1: $r^{*}(x)  =\boldsymbol{1}\left[\eta\_{h^{(2)}(x)}(x)-\eta_{h^{(1)}(x)}(x)>c\right]$. Diff-01 is based on the one-hot oracle in (3) derived as a one-sample estimator of $\eta_{h^{(i)}(x)}( x ) = \mathbb{E}\_{y \mid x}\left[ \boldsymbol{1}[y = h^{(i)}(x)] \right]$. This is a straightforward plug-in estimator of the optimal rule that requires no assumption on how close the i-th model $h^{(i)}$ is to the Bayes classifier $\mathbb{P}(y|x)$.
>
> By contrast, Diff-Prob is based on the probability oracle in (4):  $\hat{r}\_{{\rm prob}}(x) = \boldsymbol{1}\left[ p_y^{(2)}(x) - p_{y}^{(1)}(x) > c\right]$. Deriving this rule assumes that $\mathbb{P}(h^{(i)}(x) = y \mid x)$ is close to $p_y^{(i)}(x)$ for $i \in \{1,2\}$. This may not hold in practice.
>
> > theoretical analysis for the Bayes optimal deferral rule for a cascade of K=2
>
> We note that Lemma F.1 presents the optimal deferral rule for cascades of $K>2$ models. This supplements Proposition 3.1 in the main text that characterises the optimal deferral rule for cascades of $K=2$ models.

---

> > ### Comment · Reviewer_kqwH · 2023-08-16
> >
> > I have read the rebuttal. My comments were addressed by the authors.

---

### Official Review · Reviewer_UXcD · 2023-07-31

**Soundness:** 4 excellent
**Presentation:** 4 excellent
**Contribution:** 3 good
**Rating:** 7
**Confidence:** 4

**Summary:**

In this paper, the authors study the problem of confidence-based cascade deferral for the classification task. They first proposed the formulation of Bayes optimal deferral rule, which relies on both the base model and its successive model, and then proposed several baselines that works by mimicking the Bayes optimal rule. They further analyzed confidence-based methods and proposed the condition for their consistency. Given the fact that these baselines' computational costs are not acceptable, they turn to propose a post-hoc method guided by the baselines and analyzed its consistency under different conditions. Experimental results validated the performance of the proposed method.

**Strengths:**

1. The authors studied the optimality condition for cascade deferral rule and show the formulation of the Bayes optimal rule that relies on the confidence score of two models, which takes an intuitive form that implies the limitation of confidence-based deferral.

2. The authors gave several baselines to approximate the Bayes optimal rule, and further proposed the post-hoc versions for them to resolve the computational issues.

3. Theoretical analyses are conducted for both the confidence-based method and post-hoc deferral rule under various conditions.

**Weaknesses:**

1. Though the connection between this work and learning to defer/ classification with rejection is revealed, it is not thoroughly investigated in my opinion. In fact, learning to defer and classification with rejection are shown to be equivalent to the ordinary multi-class classification problem [1, 2]. In the post-hoc regime, I think it is worth trying to reduce the training of the cascade deferral rule to a multi-class classification problem by setting the maximum confidence of each model as a pseudo posterior probability.

2. As stated in the limitation section, the analyses are conducted on a case-by-case basis, indicating that there is still progress needed to develop a consistent and computationally friendly deferral rule. While the experimental results suggest that the proposed method is comparable, it is still a matter worth addressing. Conducting a finite-sample analysis could significantly alleviate this concern.

[1]. Mohammad-Amin Charusaie, Hussein Mozannar, David A. Sontag, Samira Samadi. Sample Efficient Learning of Predictors that Complement Humans. ICML 2022.

[2]. Yuzhou Cao, Tianchi Cai, Lei Feng, Lihong Gu, Jinjie Gu, Bo An, Gang Niu, Masashi Sugiyama. Generalizing Consistent Multi-Class Classification with Rejection to be Compatible with Arbitrary Losses. NeurIPS 2022.

**Questions:**

Please see the Weaknesses section.

**Limitations:**

The limitations are adequately analyzed in Appendix G.

---

> ### Author Rebuttal · Authors · 2023-08-10
>
>
> > The analyses are conducted on a case-by-case basis. There is still progress needed to develop a consistent and computationally friendly deferral rule.
>
> We reiterate that by Proposition 3.1, the Bayes-optimal deferral rule is based on thresholding $s^*( x ) = \eta_{h^{(1)}(x)}(x) - \eta_{h^{(2)}(x)}(x)$. By Lemma 4.1, confidence based deferral is optimal if $\eta_{h^{(1)}(x)}(x)$ produces the same order over input instances compared to $s^*( x )$. This one condition characterises the optimality of the confidence based deferral and thus can be seen as a unified (population level) analysis.
>
> The three cases presented in Section 4.1 (i.e., specialist setting, label noise, distribution shift) are intuitive problem instances where the confidence-based deferral is not optimal. Note that in experiments on all these three settings, the same set of post-hoc approaches and the same training recipe are used (see Section C.2). In all cases, the post-hoc model is a small MLP composed of no more than 100 hidden units. The overhead in the inference cost from calling the post-hoc model is negligible, and thus we believe this approach is computationally-friendly.
>
>
> > Conducting a finite-sample analysis could significantly alleviate this concern.
>
> We agree with the reviewer that conducting a finite-sample analysis is of interest. As we demonstrate next, **the Diff-01 method readily admits standard generalisation bounds.**
>
> ### Finite sample analysis
>
>
> Recall that for given base classifier $h^{(1)}$ and $h^{(2)}$ and deferral cost $c$, the optimal deferral rule takes the form
> $r^*(x) = 1[g^*(x) > c]$,
> where
> $g^*(x) = \eta_{h^{(2)}(x)}(x)-\eta_{h^{(1)}(x)}(x)$.
> We seek to approximate this optimal rule by constructing samples $S_{\rm{val}} := \{ ( x_i, z_i) \}$, where the labels $z$ are chosen such that $\mathbb{E}[ z | x] = g^*(x)$. We then pick a scorer $\hat{g}$ from a hypothesis class $\mathcal{G}$ that minimizes the empirical average squared loss on this sample:
> $$\hat{R}\_{\rm sq}(g) \,= \frac{1}{| S\_{\rm{val}} |} \sum\_{( x_i, z_i ) \in S\_{\rm val}} ( z_i - g( x_i ) )^2,$$
> and construct a deferral rule  $\hat{r}(x) = 1[\hat{g}(x) > c]$.
>
> We have the following result.
>
> **Proposition**
> Let $\mathcal{N}(\mathcal{G}, \epsilon)$ denote the covering number for $\mathcal{G}$ with the $\infty$-norm.  Suppose for any $g \in \mathcal{G}$, the squared loss $(z - g(x))^2 \leq B, \forall (x, z)$.
> Fix $\delta \in (0,1)$. Then  with probability at least $1 - \delta$ over draw of $S_{\rm val}$,
>
> $$R(\hat{r};h^{(1)},h^{(2)}) - R(r^*;h^{(1)},h^{(2)})
> \leq
>     \mathbb{E}\_x [ (\tilde{g}(x)  -  g^*(x))^2 ]  +
>  4\cdot\inf\_{\epsilon > 0}
>         \epsilon +
>         B\sqrt{
>             \frac{ 2\cdot\mathcal{N}(\mathcal{G}, \epsilon) }{|S\_{\rm val}|}
>         }
>     +
>     \mathcal{O} (\sqrt{\frac{\log(1/ \delta)}{|S\_{\rm val}|}})
> $$
>
> In the above finite-sample bound, the first term on the right-hand side (RHS) is an  approximation error quantifying the distance between the best possible model $\tilde{g}$ in the class $\mathcal{G}$ to the optimal function $g^*$. The second and the third terms on RHS can be interpreted as an estimation error.
>
>
> > Though the connection between this work and learning to defer/ classification with rejection is revealed, it is not thoroughly investigated
>
> **The reviewer is completely correct that there are close links between model cascading and learning to defer.** However, as this is not the primary focus of the work, in the interest of space we “deferred” a detailed discussion of this topic to the cited work of [Narasimhan et al., 2022]. We are happy to add citations to the referenced works from the learning to defer literature.

---

> > ### Comment · Reviewer_UXcD · 2023-08-20
> >
> > Thanks for the authors' detailed response. My concerns are all addressed, and I will keep the score.

---

### Author Rebuttal · Authors · 2023-08-10

We thank all the reviewers for constructive comments. We will revise our submission accordingly. In what follows, we clarify common points raised by multiple reviewers. *Please note that we have also attached a PDF to this rebuttal.*


> Meaning of “produce the same order” in Lemma 4.1 [Reviewer kqwH, Reviewer ahRx]

**We will make this statement more clear in the revised version.** First, we recall the statement of Lemma 4.1:

__Lemma 4.1__ The deferral rule $r_{\mathrm{conf}}$ produces the same deferral curve as the Bayes-optimal rule (2) __if__ $\eta_{h^{(1)}(x)}(x)$ and $\eta_{h^{(1)}(x)} (x) − \eta_{h^{(2)}(x)} (x)$ produce the same ordering over instances $x \in \mathcal{X}$.

For brevity, we write $\eta_{i}(x)$ for $\eta_{h^{(i)}(x)}(x)$.
* Define real-valued random variables $ z := \eta_{1}(x)$, and $z^* := \eta_{1}(x)-\eta_{2}(x)$
where $x \sim \mathbb{P}_{x}$.
* Let $\gamma_{\alpha},\gamma_{\alpha}^*$ be the $\alpha$-quantile of the distributions of $z$ and $z^*$, respectively.

By “produce the same ordering over instances $x \in \mathcal{X}$, we precisely mean that for any $x \in \mathcal{X}$ and $\alpha \in [0, 1]$, $\eta_1(x) < \gamma_\alpha \iff \eta_1(x) - \eta_2(x) < \gamma^*_\alpha$. We use this definition in the proof.

---

### Decision · Program_Chairs · 2023-09-21

**Decision:**

Accept (poster)

**Comment:**

This paper analyzes classifier cascades through the lens of learning-to-defer/reject, studying when the sensible but naive rule of confidence-based deferral would (not) suffice.  Failure modes include when the downstream classifier is a narrow expert and when there is distribution shift.  The reviewers are unanimously in favor of acceptance, agreeing that the work provides a fresh perspective on classifier cascades.  The most significant criticisms are the issue of what is meant by ordering (Reviewer kqwH, Reviewer ahRx) and the practical considerations highlighted by Reviewer ahRx (or rather, the presentation of the issue).  The authors should address these concerns / suggestions in the camera-ready version.